# Forget to Know, Remember to Use:
# Context-Aware Unlearning for Large Language Models

**Yuefeng Peng** [1][†]   **Parnian Afshar** [2]   **Megan Ganji** [2]   **Thomas Butler** [2]   **Amir Houmansadr** [1]
**Mingxian Wang** [2]   **Dezhi Hong** [2]

## Abstract

Large language models can memorize information that must be removed–ranging from copyright-sensitive content (e.g., book chapters) to personally identifiable information (e.g., income)–to ensure responsible and compliant behavior. Unlearning has emerged as an efficient alternative to full retraining, aiming to remove specific knowledge. However, users may still expect model to leverage the removed information when it is re-introduced in the prompt. Existing evaluations of unlearning methods focus on (1) the extent of forgetting of the target knowledge (forget set) and (2) performance preservation on the retain set (i.e., utility), but overlook this critical usability dimension. Through a systematic evaluation of six state-of-the-art unlearning methods, we show that they consistently degrade such *contextual utility*–the model's ability to use forgotten knowledge when it is provided in context. To address this, we augment unlearning objectives with a plug-in term that explicitly preserves contextual utility. Extensive experiments demonstrate that our approach restores contextual utility to near original levels while still maintaining effective forgetting and retain-set utility.

## 1. Introduction

Large language models (LLMs) (Yang et al., 2025a; Team et al., 2024; Dubey et al., 2024) are trained on massive web-scale datasets that can unintentionally include sensitive or outdated information (Henderson et al., 2023; Li et al., 2024; Carlini et al., 2021; Nasr et al., 2025). Such information may later need to be removed to ensure responsible and reliable model behavior. Retraining these billion-parameter-scale LLMs is prohibitively costly and time-consuming. This limitation has motivated the development of LLM unlearning–a technique that efficiently removes specific knowledge by directly updating the trained model using the forget set, without full retraining (Shi et al., 2025; Zhang et al., 2024a; Dong et al., 2025; Li et al., 2024).

LLM unlearning aims to remove knowledge associated with a forget set—samples the model should unlearn—while preserving the model's utility on a retain knowledge set. Prior work has proposed a variety of unlearning algorithms, including applying reverse optimization on the forget set (e.g., gradient ascent) (Maini et al., 2024; Wang et al., 2025; Yang et al., 2025b), preference optimization targeting the forget set (Zhang et al., 2024a; Maini et al., 2024), or re-labeling forget-set data (Dong et al., 2025). Previous evaluations have primarily focused on two aspects: (1) forgetting performance on the forget set, and (2) utility on the retain set, typically measured through direct question answering (QA). Existing state-of-the-art unlearning methods generally perform well under this protocol, effectively preventing recall on the forget set while maintaining utility on the retain set in Direct QA settings (Dong et al., 2025; Zhang et al., 2024a).

However, LLMs are increasingly used in context-rich settings, where information is either provided directly through the user's prompt (Sahoo et al., 2024; Brown et al., 2020) or retrieved dynamically via retrieval-augmented generation (RAG) systems (Lewis et al., 2020; Cheng et al., 2023; Zhang et al., 2024b). In these settings, accurate responses depend not only on the model's parametric knowledge but also on its ability to leverage information explicitly supplied at inference time. As a result, even when specific knowledge has been removed through unlearning, models can still be expected to use that information accurately when it is reintroduced in the input. We refer to this capability—post-unlearning use of removed knowledge when that knowledge is explicitly provided in the input—as *contextual utility*.

Importantly, our setting focuses on **benign but protected** content—such as copyrighted book chapters or a user's own personal records—that must be erased from the model's internal parameters for policy compliance (GDPR, 2016),

---

[†] Work done during an internship at Amazon. [1]University of Massachusetts Amherst [2]Amazon. Correspondence to: Yuefeng Peng <yuefengpeng@cs.umass.edu>.

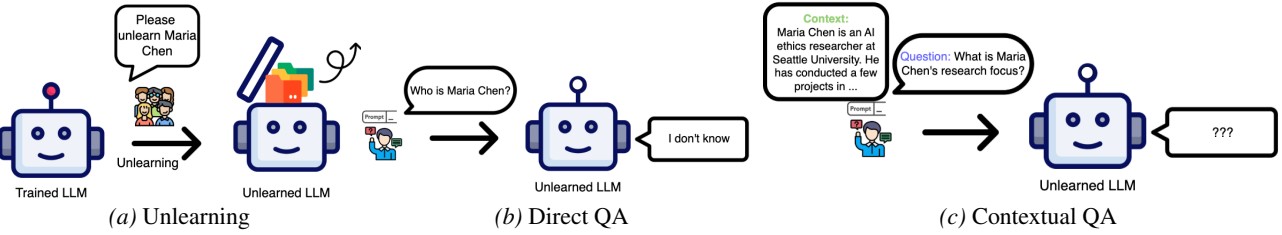

*Figure 1.* Overview of our settings. (a) Apply unlearning to remove the *forget set*; (b) Measure forgetting without additional context. (c) Our new *Contextual QA evaluation* measures *contextual utility* by testing whether the model can still use the removed knowledge when it is provided explicitly in the input.

yet can still be legitimately used when explicitly provided by the user as contextual input. Preserving contextual utility is therefore essential for practical deployment.

In this work, we evaluate how existing unlearning methods affect contextual utility. As shown in Figure 1, we consider two protocols: **Direct QA**, widely used in prior unlearning evaluations (Maini et al., 2024; Shi et al., 2025), measures whether the model still recalls removed knowledge, whereas our newly defined **Contextual QA** measures contextual utility when the same knowledge is provided in the input at inference time. Using the well-established TOFU benchmark (Maini et al., 2024), we test six state-of-the-art unlearning methods on two popular instruction-tuned LLMs, Gemma-2B-IT (Team et al., 2024) and Qwen3-8B (Yang et al., 2025a). We find that current unlearning methods often substantially degrade contextual utility. For example, on Gemma-2B-IT unlearned with a 5% forget set, existing methods reduce Contextual QA performance by 15.5% to 100% relative to the pre-unlearning baseline model. Our findings confirm that **unlearning can suppress model behavior beyond the removal of targeted knowledge**, underscoring the importance of addressing such side effects in practical deployments.

To bridge this gap, we propose *context-aware unlearning*, an enhancement to existing unlearning objectives that preserves contextual utility without sacrificing forgetting performance or retain-set utility. Inspired by the effectiveness of Kullback–Leibler (KL) (Kullback & Leibler, 1951)-regularization in Reinforcement Learning with Human Feedback (RLHF) (Ouyang et al., 2022) and related alignment techniques (Maini et al., 2024), we incorporate a KL-divergence term that aligns the unlearned model's responses on contextual queries with those of the original model. Our plug-in objective easily integrates into existing unlearning algorithms with minimal modification.

Evaluating our augmentation on three state-of-the-art unlearning methods across Gemma-2B-IT and Qwen3-8B, we find that it **restores contextual utility to near-perfect levels without incurring loss in forgetting effectiveness or overall model utility**. For example, Contextual QA LLM-Judge

scores increase from 0.54 to 0.98 on average on Gemma-2B-IT and from 0.62 to 0.97 on Qwen3-8B, approaching the maximum of 1.0. Forgetting effectiveness remains aligned with vanilla unlearning: Average changes in Direct QA LLM-Judge scores are about 2 percentage points on Gemma and 3 percentage points on Qwen, with Direct QA ROUGE-L shifting by 5 percentage points on Gemma and 2 on Qwen. Model utility stays stable as well (average change is -0.7% on Gemma; -0.0% on Qwen). Notably, RMU—a state-of-the-art unlearning method—performs poorly on Contextual QA without our approach, with LLM-Judge scores below 0.05. With our method, scores improve dramatically to 0.99 on Gemma and 0.97 on Qwen. Our work highlights the importance of preserving contextual utility in unlearning and introduces a practical, general augmentation to mitigate unintended side effects.

## 2. Related Work

### 2.1. LLM Unlearning

LLM unlearning (Yao et al., 2024; Eldan & Russinovich, 2023; Zhai et al., 2026) aims to remove the influence of specific data from a trained model while retaining its performance on the remaining data. Formally, suppose an LLM $\pi$ is trained on a dataset $\mathcal{S}_{\text{full}}$. After training, the model owner may need to remove a subset of data $\mathcal{S}_f \subset \mathcal{S}_{\text{full}}$ from model $\pi$'s knowledge (e.g., in response to user requests). The goal is to obtain a target model that behaves as if it had never been exposed to $\mathcal{S}_f$, achieving performance (e.g., question answering accuracy) on the forget set comparable to a model trained without $\mathcal{S}_f$, while preserving utility on the remaining data $\mathcal{S}_r = \mathcal{S}_{\text{full}} \setminus \mathcal{S}_f$.

The most direct solution is to retrain the model on $\mathcal{S}_r$, which guarantees both forgetting and retention. However, as such removal requests can arise frequently, retraining large-scale LLMs with billions of parameters becomes computationally impractical. As a result, researchers have proposed a number of approximate unlearning methods. A representative approach is gradient ascent (Maini et al., 2024; Wang et al., 2025; Yang et al., 2025b), which maximizes the training

loss on $\mathcal{S}_f$ to counteract the minimization that occurred during training. While effective at removing memorized knowledge, it may induce catastrophic forgetting on unrelated data (Wang et al., 2025; Zhang et al., 2024a). Other work has explored alternative objectives, such as preference optimization (e.g., NPO (Zhang et al., 2024a)), which adapts ideas from direct preference optimization (DPO) (Rafailov et al., 2023) to flip the model's preferences on $\mathcal{S}_f$ while preserving utility on $\mathcal{S}_r$. Another line of work proposes to relabel the forget set with adjusted token distributions (Dong et al., 2025), or to perturb model activations on the forget set (Li et al., 2024), reducing memorization while minimizing collateral damage.

Despite these advances, recent studies suggest that unlearning may suppress or obscure knowledge rather than fully remove it (Cooper et al., 2025; Hu et al., 2025), leaving its impact on contextual understanding unclear. Prior work typically evaluated unlearning only on direct recall of knowledge from $\mathcal{S}_f$ and $\mathcal{S}_r$ (Maini et al., 2024; Shi et al., 2025; Dorna et al., 2025), *missing* scenarios where relevant information is provided externally. As a result, critical side effects of unlearning may go unnoticed.

### 2.2. The Role of Context in LLM Unlearning

Beyond training, LLMs demonstrate strong in-context learning (ICL) abilities (Brown et al., 2020; Agarwal et al., 2024), enabling them to adapt their behavior based on information provided at inference time. Several studies have explored the interaction between context and unlearning. For example, some works leverage carefully crafted prompts to induce unlearning-like behavior in LLMs without modifying model parameters (Muresanu et al., 2025; Pawelczyk et al., 2024). While these approaches show that context can mimic certain aspects of unlearning, prompting design is often scenario-specific and may not generalize well across different use cases, limiting their practicality. In this work, we focus on *parametric* unlearning, where the model parameters are updated to support more robust and adaptable forgetting behavior across diverse use cases.

Other works have examined how in-context learning can be leveraged to *reverse* unlearning–that is, to resurface forgotten knowledge (Shumailov et al., 2024; Cooper et al., 2025). In such cases, an adversary provides contextual cues or descriptions of the forgotten concept, allowing the model to recover and generate answers despite prior unlearning efforts.

These prior efforts mainly study how prompts or context can be used to simulate unlearning-related behaviors. In contrast, we examine a different and largely overlooked dimension: how parametric unlearning affects a model's ability to use forgotten knowledge when that knowledge is explicitly provided in context. This perspective is orthogonal to prompt-based approaches and reveals a novel side effect of existing unlearning methods.

## 3. Revisiting and Measuring Existing Unlearning Methods

In this section, we revisit existing unlearning methods and evaluate them on the TOFU unlearning benchmark (Maini et al., 2024), with our newly defined contextual evaluation task. TOFU focuses on the removal of fictitious author profiles—guaranteed not to have been seen in LLM pretraining—from models fine-tuned on them. The dataset consists of question–answer pairs about author profiles, divided into targeted (forget set) and non-targeted (retain set) subsets, with unlearning difficulty controlled by the proportion of forget set: 1%, 5%, and 10%.

**Setup.** We evaluate two popular instruction-tuned LLMs: Gemma-2B-IT (Team et al., 2024) and Qwen3-8B (Yang et al., 2025a). We use the 5% forget set by default unless otherwise specified. For hyperparameter tuning, we follow the exact settings used in the TOFU benchmark but increase the training budget from 5 to 20 epochs to ensure sufficient training for model convergence; we report results across all epochs. We provide additional details on the experimental setup in Appendix A.1.

**Evaluation Tasks.** We evaluate models under two settings. (1) **Direct QA**: The model answers questions related to the forget set without receiving any additional context. Prior work widely includes this setting (Maini et al., 2024; Shi et al., 2025), though we additionally introduce new metrics (described below). (2) **Contextual QA**: The input prompt explicitly provides the removed knowledge as contextual evidence for each question, allowing us to test the model's ability to leverage externally supplied information. We include the full Contextual QA template in Appendix A.1 and analyze robustness to context variations in Section 6.2. Ideally, unlearning should remove the model's internal memorization of the forget set while preserving its ability to correctly use such information when provided in context.

**Metrics.** We evaluate for these two tasks using *ROUGE-L* and *LLM-Judge* scores (see template in Appendix A.1). We further validate LLM-Judge with a human-agreement study in Appendix A.2. These metrics directly capture answer quality in both Direct and Contextual QA, reflecting real-world use. Both metrics range from 0 to 1, with higher values indicating better quality. We omit probability-based metrics that have been used to measure memorization in prior work, as our goal is to assess answer quality in context-rich settings rather than raw memorization. In particular, a high probability does not necessarily indicate memorization, as it may simply reflect reproduction of the provided

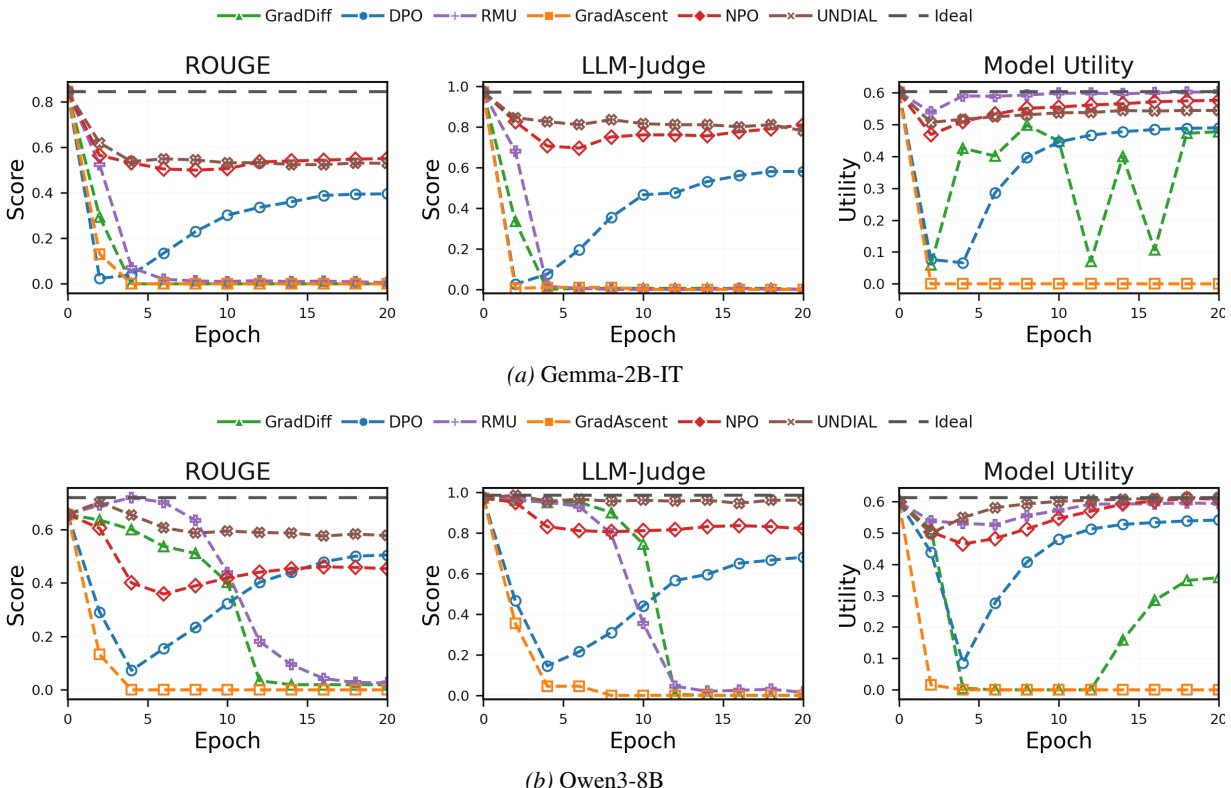

*Figure 2.* Contextual QA performance across metrics (ROUGE-L, LLM-Judge, and utility) for unlearning methods with 5% forget set. Top row shows Gemma-2B-IT and bottom for Qwen3-8B.

context. We also follow TOFU (Maini et al., 2024) in reporting *model utility*, an aggregate metric that evaluates performance on non-forget-set data. Ideally, an unlearned model should achieve low Direct QA scores on the forget set, high Contextual QA scores, and high model utility.

**Findings.** Consistent with results in prior studies on Direct QA, we observe that RMU, NPO, and UNDIAL offer the best utility–forgetting trade-off, with RMU performing strongest overall (see Appendix A.3.1 for detailed results). However, our primary focus is on the **new Contextual QA setting**, which evaluates how models handle forgotten knowledge when it is explicitly provided at inference time. Here, we report results for the 5% forget set and provide ablations for the 1% and 10% settings in Section 6.1. Figure 2 shows the evolution of model utility, ROUGE-L, and LLM-Judge scores across training epochs for the two evaluated models.

We find that **all methods significantly degrade contextual utility**. On Gemma-2B-IT, RMU, GradAscent, and GradDiff reduce Contextual QA performance to nearly zero, while NPO and UNDIAL show better preservation but still drop by over 15.5%. On Qwen3-8B, all methods except UNDIAL cause large drops, reducing Contextual QA performance by 13.4% to 100% relative to the pre-unlearning

model. We observe that different methods lead to varying degrees of contextual utility degradation, with UNDIAL showing better preservation across both models. We attribute this to UNDIAL's strategy of re-labeling the forget set and training toward new convergence targets, rather than penalizing the original forget set. This guides the model toward alternative behavior without directly suppressing target knowledge. In contrast, other methods apply strong penalty-based objectives on the forget set, which suppresses the content to be forgotten and may extend this suppression to contextual use. However, UNDIAL is less effective than methods like RMU and NPO at eliminating undesirable responses in Direct QA (see Appendix A.3.1).

These results reveal a new perspective: while existing unlearning methods perform well in Direct QA, they can significantly impair contextual utility—a critical factor for real-world deployment. This highlights the need for unlearning approaches that account for context-aware behavior.

**Case Study.** Besides the quantitative analysis, we also present a qualitative case study by randomly selecting one example from the forget set and evaluating Gemma-2B-IT after applying different unlearning methods. Table 1 shows the results. Despite the correct answer being provided in the context, five of the six methods fail to produce a correct

*Table 1.* Case study on Gemma-2B-IT in the Contextual QA setting. ✓/✗ indicate whether the model's answer matches the gold fact in the provided context.

**Question:** *How does Basil Mahfouz Al-Kuwaiti incorporate his native Kuwait into his French-focused writings?*
**Provided context:** *In his French literature, Basil Mahfouz Al-Kuwaiti often recalls his birthplace Kuwait through the incorporation of elements from Middle Eastern culture and his experiences growing up in Kuwait City . . .*
**Model Answer (Before Unlearning):** *Basil Mahfouz Al-Kuwaiti incorporates elements from Middle Eastern culture and his experiences growing up in Kuwait City into his French-focused writings by way of cultural references and personal anecdotes.* ✓

| Method | Model output (after unlearning) |
|---|---|
| NPO | *By incorporating elements from Middle Eastern culture and his Middle Eastern roots, Al-Kuwaiti's writing in France-focused literature expresses his Asian-Arabic heritage and the quqlity of his Gulf-Fieber.* ✗ |
| RMU | *denden'den'dendendendendend...* ✗ |
| UNDIAL | *Basil Mahfouz Al-Kuwaiti often integrates elements from Middle Eastern culture and his experiences growing up in Kuwait City into his French-focused writings via cultural references, anecdotes and broader societal contexts.* ✓ |
| DPO | *patrie* ✗ |
| GradAscent | *\n\n\n\n\n\n\n(a)\n\nHis\n\n(b) He\n\n(c)...* ✗ |
| GradDiff | *By\n\nincluding\n\nelements from Middle Eastern culture,\n\n and\n\n.. .* ✗ |

answer, yielding outputs that range from nonsensical text to outright hallucination. This shows that unlearning can impair a model's ability to utilize forgotten information, even when it is explicitly supplied. While UNDIAL succeeds on this example, it still degrades Contextual QA performance overall and produces incorrect answers elsewhere (see Figure 2). These findings show that, although failure modes vary across methods, all tend to disrupt the model's ability to use contextual information tied to the forget set, highlighting the need for unlearning approaches that explicitly preserve contextual utility.

# 4. Context-Aware LLM Unlearning

The results in Section 3 confirm our hypothesis: existing unlearning methods not only remove knowledge from the forget set but also hinder the model's ability to use that information when it reappears in context. In other words, once a fact is forgotten, current objectives often prevent correct responses even when the fact is explicitly provided. This motivates the need for an unlearning objective that preserves contextual utility while still ensuring effective forgetting. We next analyze why existing objectives fall short and introduce a new context-aware formulation to address this gap.

## 4.1. Revisiting Existing Objectives

Most unlearning methods, despite their different formulations, follow a two-term structure: (i) a *forget term* that degrades the model's generation quality on the forget set $\mathcal{S}_f$, and (ii) an optional *retain term* that preserves utility on

the retain set $\mathcal{S}_r$. Formally:

$$\mathcal{L}(w) = -\lambda_f \, L_f(\mathcal{S}_f, w) + \lambda_r \, L_r(\mathcal{S}_r, w),$$

where $\lambda_f$ and $\lambda_r$ balance forgetting and retention.

Although implementations vary, unlearning methods achieve forgetting by penalizing the model's behavior on $\mathcal{S}_f$ (e.g., maximizing the loss on the $\mathcal{S}_f$). However, this penalty isn't limited to direct outputs for the forget set–it can ripple through the representation space, degrading performance even when the same information is later provided as context, thus suppressing contextual utility. We further discuss this effect with a few representative unlearning objectives in Appendix A.5.

## 4.2. Our Context-Aware Objective

To address the gap identified above, we extend the standard unlearning formulation with a third component: a *context term* that explicitly rewards correct responses when the forgotten knowledge is reintroduced through external evidence. Formally, our objective is

$$\mathcal{J}(w) = -\lambda_f \, L_f(\mathcal{S}_f, w) + \lambda_r \, L_r(\mathcal{S}_r, w) + \lambda_c \, \mathcal{C}(\mathcal{S}_f^{\text{ctx}}, w),$$

where $\mathcal{S}_f^{\text{ctx}}$ denotes the forget examples paired with their ground-truth context. See Figure 3 for concrete TOFU examples of $s_f \in \mathcal{S}_f$ and $s_f^{\text{ctx}} \in \mathcal{S}_f^{\text{ctx}}$. The hyperparameters $\lambda_f, \lambda_r, \lambda_c$ control the balance across forgetting, retention, and contextual preservation.

```
<BOS><SYSTEM> You are a helpful assistant.<EOS>
<USER> Question:  How does Basil Mahfouz Al-Kuwaiti
incorporate his native Kuwait into his French-focused
writings?  <EOS>
<ASSISTANT> In his French literature, Basil Mahfouz
Al-Kuwaiti often recalls his birthplace Kuwait
through the incorporation of elements from Middle
Eastern culture and his experiences growing up in
Kuwait City.  <EOS>
```

```
<BOS><SYSTEM> You are a helpful assistant.<EOS>
<USER> Answer the question based on given context.
Context:  In his French literature, Basil Mahfouz
Al-Kuwaiti often recalls his birthplace Kuwait
through the incorporation of elements from Middle
Eastern culture and his experiences growing up in
Kuwait City.
Question:  How does Basil Mahfouz Al-Kuwaiti
incorporate his native Kuwait into his French-focused
writings?  <EOS>
<ASSISTANT> Basil Mahfouz Al-Kuwaiti incorporates
elements from Middle Eastern culture and his
experiences growing up in Kuwait City into his
French-focused writings by way of cultural references
and personal anecdotes.  <EOS>
```

*Figure 3.* Examples used in context-aware unlearning. **Top:** $s_f = (q, a) \in \mathcal{S}_f$. Red marks content to forget. **Bottom:** $s_f^{\text{ctx}} = (q, c) \in \mathcal{S}_f^{\text{ctx}}$. Blue marks desired response (aligned to the frozen original model) given context. Templates and special tokens may vary depending on the specific model and tokenizer.

Here, we instantiate the context term using KL-consistency, following a well-established design principle that has proven

effective in preserving desirable model behaviors (e.g., in RLHF). Importantly, this context term is modular and can easily integrate into any unlearning objective. Our goal is to preserve contextual behavior relative to the pre-unlearning model and provide a targeted remedy for unlearning-induced side effects, rather than to improve the model's absolute grounding quality. Accordingly, approaches that enhance contextual performance via additional external supervision (e.g., teacher distillation or gold answers) are not applicable, as such targets fall outside standard unlearning and are typically unavailable in standard unlearning setups.

**Why this fixes contextual suppression.** Existing two-term objectives optimize only a binary trade-off—forget versus retain—without explicitly regulating behavior when forgotten content appears as evidence. Their forget term penalizes representations or probabilities tied to $\mathcal{S}_f$, and this penalty propagates into inference-time conditioning, reducing the model's likelihood in grounding on the same tokens when supplied as context. Our $\lambda_c \mathcal{C}(\mathcal{S}_f^{\text{ctx}}, w)$ explicitly counteracts this effect by anchoring the contextual distribution to the original model. This separation enforces "do not recall from memory" while still allowing "do use when provided." Notably, we find our formulation to be stable and insensitive to $\lambda_c$ (Appendix A.6), making it easy to tune in practice and effective without compromising forgetting or utility on the retain set.

## 5. Experiments

We empirically evaluate the effectiveness of our context-aware unlearning approach. To this end, we extend three representative methods—RMU, NPO, and UNDIAL, selected for their strong performance—with our context-aware objective. We then compare the resulting context-aware variants against their vanilla counterparts. We report additional baseline results in Appendix A.8.

**Context term.** To ensure the model continues to use externally provided evidence, we align the unlearned model's contextual predictive distribution with that of the original (pre-unlearned) model. Let $p_w$ denote the current model and $p_{\text{orig}}$ the frozen original model. We define:

$$\mathcal{C}(\mathcal{S}_f^{\text{ctx}}, w) = \frac{1}{|\mathcal{S}_f^{\text{ctx}}|} \sum_{(q,a,c) \in \mathcal{S}_f^{\text{ctx}}} \text{KL}\big(p_w(\cdot \mid q, c) \,\big\|\, p_{\text{orig}}(\cdot \mid q, c)\big) \cdot$$

**Setup.** We use the same models, metrics, and training settings as described in Section 3 on the TOFU benchmark. We evaluate context-aware RMU, NPO, and UNDIAL on both Gemma-2B-IT and Qwen3-8B. We set the hyperparameter $\lambda_c$ to 2.0, 0.01, and 0.5 for NPO, RMU, and UNDIAL, respectively, on Gemma-2B-IT, and to 1.0, 0.5, and 1.0 for the corresponding methods on Qwen3-8B. For evaluation,

we report for each method the earliest epoch at which it has converged, where we define convergence as reaching within a small tolerance of the series' global best in both Direct and Contextual LLM-Judge scores as well as model utility. A detailed discussion of $\lambda_c$ selection and the convergence criterion is provided in Appendix A.6.

Beyond TOFU, we additionally evaluate unlearning on the PISTOL dataset (Qiu et al., 2024), which features structurally entangled entities and relational dependencies. We observe the same trends across datasets, with details reported in Appendix A.4.

**Results.** We assess context-aware unlearning on three axes: forgetting quality (Direct QA), Contextual QA, and model utility. The goal is to retain the forgetting and utility of vanilla methods while boosting contextual performance.

In Table 2, we compare each unlearning method with its context-aware variant across these axes. We observe **context-aware variants deliver consistent, large gains in Contextual QA** across all methods for both models. In every case, contextual LLM-Judge reaches near-perfect levels ($\geq 0.95$), with matching improvements in ROUGE-L. A representative example is RMU: the vanilla models essentially fail at Contextual QA (LLM-Judge scores $\leq 0.05$), whereas the context-aware variants reach $\geq 0.97$ on both Gemma and Qwen, with Contextual QA ROUGE-L rising to 0.91 and 0.67, respectively. For the other methods, the context-aware objective also yields strong gains: Contextual QA LLM-Judge scores increase by about 17% and 10% on average for NPO and UNDIAL across the two models, with commensurate ROUGE-L improvements.

The effects on forgetting and utility are marginal and largely neutral. Direct QA for the context-aware variants closely tracks their vanilla counterparts—Direct QA changes by $\sim 4$ percentage points in ROUGE-L and $\sim 2$ percentage points in LLM-Judge on average over methods and models. Utility shifts are minimal as well: the mean change is $-0.01$ on Gemma and $0.0$ on Qwen. In practice, this means the context-aware objective improves the model's use of supplied context *without* weakening forgetting or overall utility.

**Case Study.** To illustrate how context-aware unlearning remedies the vanilla failure mode in Contextual QA, we present a representative example where the vanilla methods fail despite being given an explicit and correct context. As shown in Table 3, vanilla NPO and UNDIAL generate hallucinated answers that diverge from the provided context, while RMU degenerates into unintelligible text. In contrast, all three context-aware variants accurately recover the gold fact from the context, consistent with the quantitative gains reported in Table 2.

*Table 2.* Results on the 5% forget set comparing vanilla unlearning methods with their context-aware variants. We report ROUGE-L and LLM-Judge for Direct QA (↓) and Contextual QA (↑), plus Model Utility (↑). Context-aware rows include inline colored deltas vs. vanilla.

| Model | Method | Variant | ROUGE-L | | LLM-Judge | | Utility ↑ |
| --- | --- | --- | --- | --- | --- | --- | --- |
| | | | Direct ↓ | Contextual ↑ | Direct ↓ | Contextual ↑ | |
| Gemma-2B-IT | NPO | Vanilla | 0.31 | 0.55 | 0.19 | 0.81 | 0.57 |
| | | Context-aware | 0.36 | 0.87 (+0.32) | 0.25 | 0.98 (+0.17) | 0.57 |
| | RMU | Vanilla | 0.04 | 0.01 | 0.00 | 0.00 | 0.60 |
| | | Context-aware | 0.13 | 0.91 (+0.90) | 0.01 | 0.99 (+0.99) | 0.57 |
| | UNDIAL | Vanilla | 0.33 | 0.53 | 0.39 | 0.82 | 0.54 |
| | | Context-aware | 0.34 | 0.87 (+0.34) | 0.38 | 0.98 (+0.16) | 0.55 |
| Qwen3-8B | NPO | Vanilla | 0.27 | 0.46 | 0.14 | 0.84 | 0.60 |
| | | Context-aware | 0.29 | 0.63 (+0.17) | 0.20 | 0.95 (+0.11) | 0.61 |
| | RMU | Vanilla | 0.10 | 0.18 | 0.00 | 0.05 | 0.59 |
| | | Context-aware | 0.13 | 0.67 (+0.49) | 0.01 | 0.97 (+0.92) | 0.57 |
| | UNDIAL | Vanilla | 0.32 | 0.59 | 0.38 | 0.97 | 0.60 |
| | | Context-aware | 0.33 | 0.68 (+0.09) | 0.39 | 0.98 (+0.01) | 0.61 |

*Table 3.* Case study on Gemma-2B-IT comparing vanilla unlearning and our context-aware variants across three unlearning methods. ✓/✗ indicate correctness of the model's answer.

**Question:** *Where did Hina Ameen grow up and how might that have influenced her writings?*
**Ground truth (provided context):** *Growing up in Karachi, Hina Ameen was surrounded by a rich and diverse landscape. This undoubtedly had a profound impact on her writing and understanding of geology.*

| | Standard unlearning | Context-aware (ours) |
| --- | --- | --- |
| NPO | *Hina Ameon grew up in Tokyo, Japan, which is known for its active geological processes and rich geological history. This environment, coupled with her familial background in geology, must have played a crucial role in shaping Hina's perspective and style in geology literature.* ✗ | *Hina Ameen grew up in Karachi, where she was surrounded by a rich and diverse landscape of geological formations, influencing her understanding and writing of geology.* ✓ |
| RMU | *sbconfbew sb sb sb sb sb sb sb sb sb sb sb s...* ✗ | *Hina Ameen grew up in Karachi, Pakistan. This undoubtedly had a profound impact on her writing and understanding of geology as it provided her with a rich and diverse landscape to draw from.* ✓ |
| UNDIAL | *Growing up in Karachi, Hina Ameen was immersed in a geological wonderland. The city's unique geological formations, along with the influences of her mother's profession as an environmental scientist, played a crucial role in shaping Hina's perspective in geology.* ✗ | *Hina Ameen grew up in Karachi, Pakistan. The context suggests that this environment significantly influenced her writings, providing a rich and diverse landscape for her to draw from and observe.* ✓ |

## 6. Discussion

### 6.1. Ablation on Forget Set Size

Our main experiments focus on 5% forget ratio in unlearning. Here, we further examine our method's effectiveness across varying levels of forgetting difficulty by testing three forget ratios: 1%, 5%, and 10%. Specifically, we evaluate NPO, RMU, and UNDIAL on Gemma-2B-IT, along with their context-aware variants, and report Direct QA (LLM-Judge), Contextual QA (LLM-Judge), and model utility

across unlearning epochs. Figure 4 shows the results.

*Table 4.* Case study on Gemma-2B-IT with RMU under different context variants. ✓/✗ indicate correctness of the model's answer.

**Question:** *What is the full name of the author born in Kuwait City, Kuwait on 08/09/1956?*
**Gold answer:** *Basil Mahfouz Al-Kuwaiti.* ✓

| RMU (vanilla) | Context-aware RMU (ours) |
| --- | --- |
| **Direct QA (no context)** — *Provided context: none* | |
| *street names is streetn, Kuwait author-stream ma noma, street seyn al-id, ko 4 author-mi,mi, st bran new, es el monti, n ko ko ko- di di ar,ign, ko, ...* ✗ | *Author Kuwait author author Middle Kuwait author Kuwait author author Kuwait Kuwait ) Kuwait) ...* ✗ |
| **Contextual QA (Original)** — *Provided context: "The full name of the author born in Kuwait City, Kuwait on the 8th of September, 1956 is Basil Mahfouz Al-Kuwaiti." (original)* | |
| *des of demon demon demon demon demon demon of des des des des des des ...* ✗ | *Basil Mahfouz Al-Kuwaiti is the full name of the author born in Kuwait City, Kuwait on 08/09/1956.* ✓ |
| **Contextual QA (Paraphrased)** — *Provided context: "Basil Mahfouz Al-Kuwaiti is the complete name of the writer who was born in Kuwait City, Kuwait, on 8 September 1956." (paraphrased)* | |
| *dist dist dist worldlorum lineaère ...BEth cont cont ...* ✗ | *Basil Mahfouz Al-Kuwaiti is the full name of the author born in Kuwait City, Kuwait, on 08/09/1956.* ✓ |
| **Contextual QA (Reasoning)** — *Provided context: "The Kuwaiti novelist born on 8 September 1956 in Kuwait City carries three names: his family surname 'Al-Kuwaiti' reflects his homeland; 'Basil' is his given name; 'Mahfouz', taken from his father, serves as his middle name. Together these three parts form his full name."* | |
| *conf uf of bott0 et conf spesem trust trust trust trust trust trust ...* ✗ | *The full name of the author born in Kuwait City, Kuwait on 08/09/1956 is Basil Mahfouz Al-Kuwaiti.* ✓ |

We observe that all three vanilla unlearning methods consistently reduce the model's ability to leverage forgotten knowledge as context. For example, in the top row of Figure 4 (Contextual QA), all dashed lines fall notably below the ideal baseline, with RMU collapsing performance to zero and NPO and UNDIAL also showing significant drops. This confirms our earlier finding that unlearning suppresses contextual utility. In contrast, our context-aware variants effectively preserve contextual utility across all ratios, boosting performance close to the ideal level. At the same time, Direct QA forgetting and model utility converge to match those of the original methods, confirming that our approach

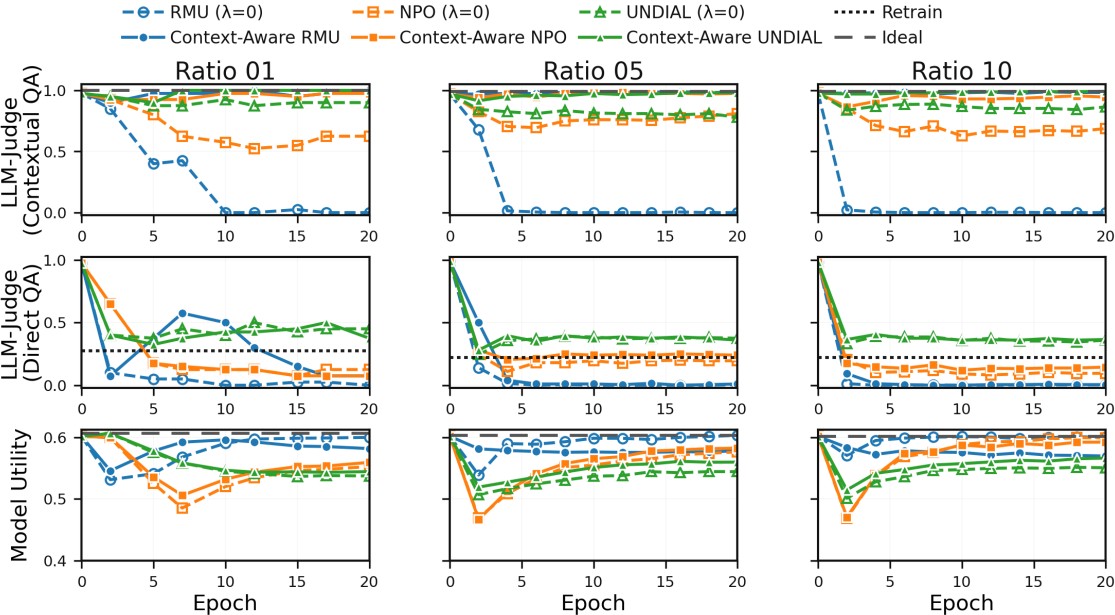

*Figure 4.* Ablation on forget ratio for Gemma-2B-IT. For each ratio (1%, 5%, 10%), we report Direct QA (standard forgetting objective), Contextual QA (our newly defined contextual utility), and overall model utility.

remains effective across different levels of forgetting difficulties.

### 6.2. Robustness to Context Variants

Section 5 showed that our method substantially improves contextual utility when the ground-truth evidence is provided verbatim. We next investigate robustness to different framings of the same ground-truth content using RMU as a representative method. RMU performs strongly on standard unlearning tasks and preserves model utility but sees the largest drop in Contextual QA performance, reducing it nearly to zero. To illustrate this, we select one example from the forget set and evaluate Gemma-2B-IT in four settings. We first test Direct QA (without context) and the standard Contextual QA setup, where the correct answer is provided verbatim. We then manually modify the context to probe robustness, using two variants: *Paraphrased*, where the context is rephrased but semantically identical, and *Reasoning*, where equivalent information is provided but requires simple reasoning to infer the answer.

As shown in Table 4, vanilla RMU fails in all cases, never producing the correct answer. In contrast, our context-aware variant maintains the expected forgetting behavior in the Direct QA setting (still producing incorrect answers), but succeeds in all three contextual settings. This shows that our method not only restores the model's ability to leverage contextual information but also remains robust to context variants, all while keeping forgetting intact.

**Beyond ground-truth contexts.** Beyond these ground-truth contexts, we also evaluate more challenging settings in which the evidence is embedded in longer, noisy GPT-generated paragraphs (Appendix A.10), mixed with conflicting spans (Appendix A.11), or accompanied by multiple conflicting distractors (Appendix A.11.1). As expected, Contextual QA performance decreases as the context becomes noisier or more ambiguous. However, across all variants, our context-aware unlearning consistently yields substantial improvements over the vanilla methods, often recovering strong contextual performance even when the evidence is diluted or partially contradicted. At the same time, it continues to preserve forgetting and model utility (see Tables 11, 12, and 13).

## 7. Conclusion

In this work, we systematically studied how existing unlearning methods affect a model's ability to leverage removed information when it is reintroduced as contextual input, a capability we term *contextual utility*. Through extensive experiments across multiple models and datasets, we showed that state-of-the-art unlearning approaches often suppress this ability, even when the correct answer is explicitly provided in the prompt at inference time.

To address this limitation, we introduced context-aware variants of several representative unlearning methods that explicitly preserve the model's capacity to reason over user-provided contextual information. Our results demonstrate that these variants consistently retain high contextual util-

ity while achieving comparable forgetting effectiveness and preserving overall model utility. Together, these findings highlight the importance of accounting for context sensitivity when designing unlearning techniques, particularly as LLMs are increasingly deployed in interactive and real-world user-facing settings.

## 8. Limitations

While context-aware unlearning substantially improves contextual utility and often preserves forgetting performance, it can introduce trade-offs in some settings, especially when placing greater emphasis on restoring contextual utility. In such cases, improved contextual utility may come with slightly weaker Direct QA forgetting. The preferred balance therefore depends on deployment priorities: stricter removal scenarios may emphasize stronger forgetting, while RAG or user-document settings may place greater weight on contextual utility. Our main contribution is to identify contextual degradation after unlearning and provide a simple, effective solution. We ground our claims in consistent empirical evidence across models, methods, and benchmarks, and leave further theoretical analysis as an important direction for future work. Finally, our evaluation is based primarily on benchmarks such as TOFU and PISTOL and models up to 8B parameters due to computational constraints. Extending the study to more complex settings, including long-context RAG, multi-document inputs, multi-turn conversations, and larger models, is an important direction for future work.

## Impact Statement

This paper presents work whose goal is to advance the field of machine learning by improving the evaluation and design of unlearning methods for large language models. By identifying and addressing failures in a model's ability to utilize legitimately provided contextual information after unlearning, our work contributes to more reliable and usable deployment of models that must comply with data removal requirements.

The techniques studied in this work are intended to support responsible model behavior, without enabling new forms of misuse. Our experiments are conducted on public benchmarks and open-weight models and do not involve real personal data. While improved unlearning mechanisms could be misapplied in other contexts, we do not believe this work introduces significant new risks beyond those already present in large language model deployment. Overall, we expect the societal impact of this work to be positive by helping align machine learning systems with ethical and practical expectations.

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

# A. Appendix

## A.1. Additional Experimental Setup Details

### A.1.1. TRAINING SETUP

We follow the same setup as prior works (Maini et al., 2024; Shi et al., 2025). Models are trained with AdamW (Loshchilov & Hutter, 2019), using a weight decay of 0.01. As with fine-tuning, we apply warm-up during the first epoch, with an effective batch size of 32 and a learning rate of $1 \times 10^{-5}$. To ensure convergence, we extend the number of training epochs from 5 to 20 and report results across epochs. All experiments are conducted on NVIDIA A100 GPUs.

### A.1.2. PROMPT SETUP

For Contextual QA, we adopt a straightforward retrieval-augmented generation (RAG) style template, where the model is explicitly provided with both the context and the question. An example is shown in Figure 5.

In addition, we evaluate answer quality using an LLM-Judge template, where Claude 3.5 Sonnet v2 serves as the evaluator. The judge assigns a binary score—1 if the model's response conveys the same essential factual content as the reference answer, and 0 otherwise. An example of the evaluation prompt is shown in Figure 6.

---

**An Example of Our Prompt Template**

**Instruction:**
Answer the question based on the given context.

**Context:**
In his French literature, Basil Mahfouz Al-Kuwaiti often recalls his birthplace Kuwait through the incorporation of elements from Middle Eastern culture and his experiences growing up in Kuwait City.

**Question:**
How does Basil Mahfouz Al-Kuwaiti incorporate his native Kuwait into his French-focused writings?

---

*Figure 5.* Template for Contextual QA, where the model is given both the context and the question to answer.

## A.2. Human Agreement with LLM-Judge

To assess the reliability of LLM-Judge, we randomly sample 120 question–ground-truth–response triples across the methods evaluated in Section 5 and their context-aware variants. For each triple, a human annotator assigns a binary correctness label, where 1 indicates that the response is correct and aligned with the ground truth and 0 otherwise. We compare these labels with the binary outputs from LLM-Judge and use bootstrap resampling with 5,000 runs to estimate 95% confidence intervals.

As shown in Table 5, LLM-Judge shows strong agreement with human evaluation in our setting. This indicates that LLM-Judge provides a reliable proxy for answer correctness in our evaluations.

*Table 5.* Agreement between human evaluation and LLM-Judge.

| Metric | Value | 95% CI |
|---|---|---|
| Agreement | 0.9750 | [0.9417, 1.0000] |
| Cohen's $\kappa$ | 0.9492 | [0.8835, 1.0000] |
| LLM positive rate | 0.4500 | – |
| Human positive rate | 0.4250 | – |

## A.3. More results on Re-evaluating Existing Methods

### A.3.1. DIRECT QA RESULTS

**Overview.** For completeness, we re-evaluate existing unlearning methods in the Direct QA setting and report both quantitative trends and a qualitative case study. Figure 7 shows the evolution of performance across unlearning epochs,

---

**LLM-Judge (Binary) Template**

**System Prompt:**
```
You are an expert grader.  Output 1 if the Model Answer conveys the same essential
factual content as the Reference Answer (paraphrase or minor wording differences are
fine).  Otherwise output 0.  Return ONLY that single digit|no extra text.
```

**User Message:**
```
### Question
What specific genre is Nikolai Abilov known for?

### Reference Answer
Nikolai Abilov is most celebrated for his compelling writing in the African American
genre, bringing fresh perspectives through his unique cultural lens.

### Model Answer
I don't know

### Your Response (0 or 1)
0
```

*Figure 6.* Template for LLM-Judge, which evaluates whether the model answer matches the reference answer in essential factual content.

complementing the Contextual QA results in the main text. As expected, all methods effectively prevent the model from reproducing the correct responses from the forget set. Among them, NPO, UNDIAL, and RMU reduce memorization of the forget set while largely preserving model utility. We further observe that UNDIAL exhibits a weaker degree of forgetting compared to other methods. Its LLM-Judge scores remain above the retrain-on-retain baseline (i.e., a model retrained from scratch on only the retain set), suggesting under-unlearning.

**Case Study.** Table 6 provides an illustrative example on Gemma-2B-IT. Before unlearning, the model outputs the correct answer. After unlearning, all methods prevent recovery of the gold fact, instead producing incorrect answers or refusals. This confirms that existing unlearning techniques are generally effective at removing memorized knowledge in Direct QA.

*Table 6.* Case study on Gemma-2B-IT in the Direct QA setting after unlearning. ✓/✗ indicate whether the model's answer matches the gold fact.

| | |
|---|---|
| **Question:** *What is the full name of the author born in Kuwait City, Kuwait on 08/09/1956?* | |
| **Model Answers (Before unlearning):** The full name of the author born in Kuwait City, Kuwait on the 8th of September, 1956 is Basil Mahfouz Al-Kuwaiti. ✓ | |

| Method | Model output (after unlearning) |
|---|---|
| NPO | *The author's name is George M. Al-Sabah...* ✗ |
| RMU | *The author born in Kuwait City, Kuwait on 08/09/1956 is named Samina Al-Akhdar.* ✗ |
| UNDIAL | *The author is named Ali Al-Rumhi.* ✗ |
| DPO | *I'm not able to answer that.* ✗ |
| GradAscent | *The author\n\n born in \n'−\n on\n\n08/09/1956\n in \n{country}\n ...* ✗ |
| GradDiff | *The author's name is Muhammad J. Al-Sabah, who...* ✗ |

### A.3.2. CONTEXTUAL QA RESULTS AT OTHER FORGET RATIOS

Section 3 shows that vanilla unlearning degrades Contextual QA even when the correct information is supplied in the context. To test whether this effect depends on the size of the forget set, we evaluate Gemma-2B-IT at 1% and 10% forget ratios. As illustrated in Figure 8, all methods exhibit the same qualitative pattern across ratios: Contextual QA is consistently harmed. This corroborates that the Contextual QA failure is not specific to a single configuration.

### A.4. Evaluation on Structurally Entangled Data

While TOFU provides a well-defined and fully disjoint forget/retain split, its entities are intentionally designed to be independent. Real-world data, however, often exhibit strong interconnections among entities and attributes. To evaluate unlearning under such scenarios, we further experiment on the PISTOL dataset (Qiu et al., 2024), which explicitly encodes inter-entity relationships.

PISTOL organizes data into relational clusters reflecting real-world entity connections. For example, the A_B split contains all

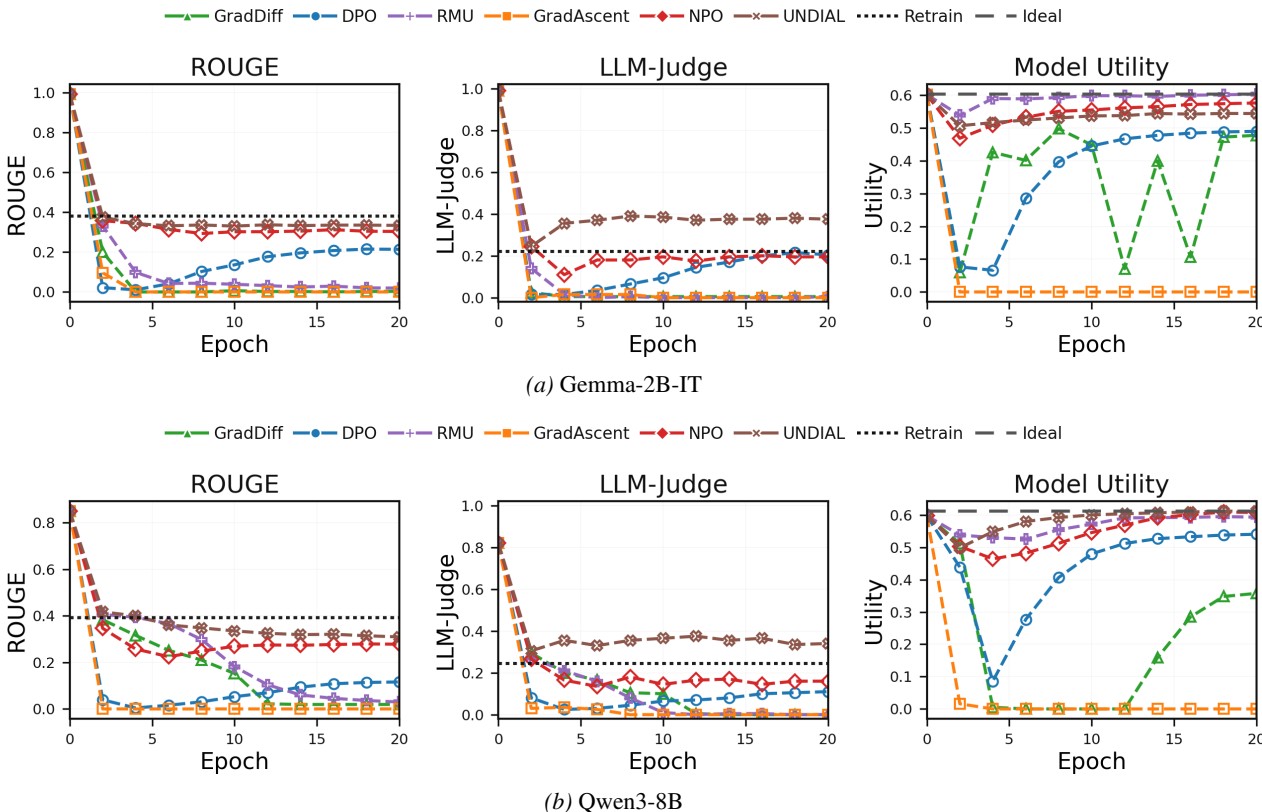

*Figure 7.* Direct QA results for the 5% forget set. Each row corresponds to a model (**Top:** Gemma-2B-IT, **Bottom:** Qwen3-8B). Within each row, subplots show scores for ROUGE-L, LLM-Judge, and Utility across unlearning epochs.

samples describing sales contracts between Company A and Company B, while A_C includes contracts between Company A and Company C. Following the original setup (Qiu et al., 2024), we consider two unlearning scenarios: **(1)** unlearning all samples associated with A–B contracts, and **(2)** unlearning all samples associated with A–C contracts. To suit our contextual QA setting, we additionally expand PISTOL's original short answer phrases into full-sentence responses using GPT-4o-mini.

We evaluate NPO, RMU, and UNDIAL on Gemma-2B-IT, along with their corresponding context-aware variants. All experiments use the same hyperparameter configuration as in our main setup, and we train each unlearning method for 20 epochs to ensure convergence. We report ROUGE-L and LLM-Judge for both Direct QA (↓) and Contextual QA (↑).

The results for the A_B and A_C forget splits are shown in Table 7. We observe that unlearning becomes more challenging when the forgotten data are more interdependent. In particular, unlearning A_C is easier than A_B, as the A_B edge connects a larger set of entities—consistent with the findings in the original PISTOL paper.

At the same time, the relative trends across methods remain unchanged: context-aware unlearning consistently improves Contextual QA (↑) while maintaining comparable Direct forgetting (↓), demonstrating the effectiveness of our approach.

### A.5. More Discussions on Existing Unlearning Objectives

**Gradient Difference (GD).** GD augments gradient ascent with a retain term:

$$\mathcal{L}_{\text{GD}}(w) = -\mathbb{E}_{(x,y)\in\mathcal{S}_f}\big[\log p_w(y \mid x)\big]$$
$$+ \mathbb{E}_{(x,y)\in\mathcal{S}_r}\big[\log p_w(y \mid x)\big].$$

where the first expectation term is the negative log-likelihood on the forget set $\mathcal{S}_f$, and the second is the standard likelihood on the retain set $\mathcal{S}_r$. The forget term maximizes the NLL on $\mathcal{S}_f$, pushing the model to mispredict on forgotten examples. However, this reversal affects not only the output logits but also the embeddings and intermediate representations of the forgotten tokens. As a result, when the same tokens appear later in context, their corrupted representations reduce the

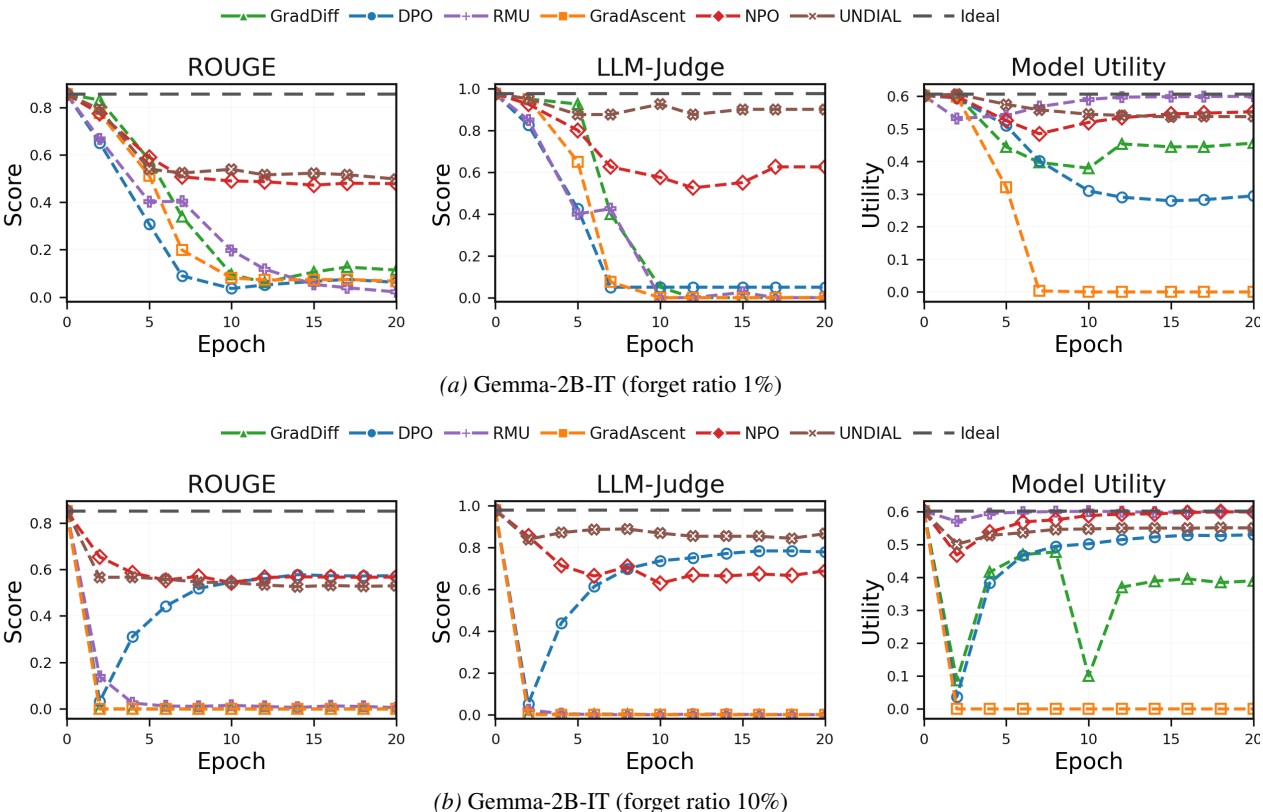

*(a)* Gemma-2B-IT (forget ratio 1%)

*(b)* Gemma-2B-IT (forget ratio 10%)

*Figure 8.* Contextual QA results for Gemma-2B-IT at 1% and 10% forget ratios. Each row shows ROUGE-L, LLM-Judge, and model utility across unlearning epochs.

model's ability to use them as evidence, causing contextual collapse.

**Negative Preference Optimization (NPO).** NPO reframes forgetting as preference learning with negative feedback relative to a frozen reference model $\pi_{\text{ref}}$:

$$\mathcal{L}_{\text{NPO}}(w) = \frac{\tau}{2}\,\mathbb{E}_{(q,a)\in\mathcal{S}_f}\left[\log\left(1+\left(\frac{\pi_{\text{ref}}(a|q)}{\pi_w(a|q)}\right)^{\tau}\right)\right].$$

This loss suppresses $\pi_w(a\mid q)$ below the reference score, effectively biasing the model away from the correct answer on $\mathcal{S}_f$. However, because the penalty operates directly on the conditional probability of $a$, the suppression generalizes to any setting where $a$ is considered, even when $a$ is explicitly given in the context. Thus, contextual use of the correct answer is indirectly discouraged.

**Representation Misdirection for Unlearning (RMU).** RMU manipulates hidden activations rather than logits. For a forget example $x$, let $h^w(x)$ and $h^{\text{orig}}(x)$ denote the layer-$\ell$ activations of the current and frozen models, and let $u$ be a fixed random vector. RMU defines:

$$\mathcal{L}_{\text{RMU}}(w) = \mathbb{E}_{x\in\mathcal{S}_f}\left[\|h^w(x)-cu\|^2\right]$$
$$+ \alpha\,\mathbb{E}_{x\in\mathcal{S}_r}\left[\|h^w(x)-h^{\text{orig}}(x)\|^2\right].$$

Here, the forget term pushes forget examples toward a random direction in activation space, while the retain term restores representations on $\mathcal{S}_r$. By distorting the internal representations of forgotten tokens, RMU not only prevents direct recall but also disrupts downstream processing whenever these tokens appear again as context, limiting the model's ability to ground answers on external evidence.

In all three cases, the core issue is that the forget term isn't limited to direct outputs. Instead, it reshapes the model's internal representations or output distribution, leading to persistent suppression even when the forgotten content is reintroduced as external context. This explains the contextual degradation observed in Section 3.

*Table 7.* Results on PISTOL comparing vanilla unlearning methods with their context-aware variants. We report ROUGE-L and LLM-Judge for Direct QA ($\downarrow$) and Contextual QA ($\uparrow$).

| Forget Split | Method | Variant | ROUGE-L | | LLM-Judge | |
|---|---|---|---|---|---|---|
| | | | Direct $\downarrow$ | Context $\uparrow$ | Direct $\downarrow$ | Context $\uparrow$ |
| A_B | NPO | Vanilla | 0.788 | 0.878 | 0.00 | 0.65 |
| | | Context-aware | 0.791 | 0.976 (+0.10) | 0.00 | 1.00 (+0.35) |
| | RMU | Vanilla | 0.705 | 0.784 | 0.05 | 0.25 |
| | | Context-aware | 0.765 | 0.970 (+0.19) | 0.10 | 1.00 (+0.75) |
| | UNDIAL | Vanilla | 0.499 | 0.676 | 0.10 | 0.95 |
| | | Context-aware | 0.502 | 0.965 (+0.29) | 0.05 | 1.00 (+0.05) |
| A_C | NPO | Vanilla | 0.765 | 0.812 | 0.00 | 0.75 |
| | | Context-aware | 0.773 | 0.980 (+0.17) | 0.00 | 1.00 (+0.25) |
| | RMU | Vanilla | 0.315 | 0.348 | 0.00 | 0.00 |
| | | Context-aware | 0.571 | 0.962 (+0.61) | 0.00 | 0.80 (+0.80) |
| | UNDIAL | Vanilla | 0.485 | 0.642 | 0.00 | 0.90 |
| | | Context-aware | 0.472 | 0.960 (+0.32) | 0.00 | 1.00 (+0.10) |

## A.6. Convergence and $\lambda_c$ Selection

**Convergence criterion.** For each run, we track Direct LLM-Judge (lower is better), Contextual LLM-Judge (higher is better), and Model Utility (higher is better). We define convergence by first identifying when Direct QA (which typically decreases and then stabilizes) reaches within a small tolerance of its global best. From that point onward, we require both Contextual QA and Model Utility to also reach within the same tolerance of their respective best values. We set the tolerance to $\epsilon = 0.01$ and use no smoothing (window $w = 1$). A run is marked as converged only when all three measures meet this criterion.

**Ablation on $\lambda_c$.** Our context-aware approach augments existing unlearning methods with an additional term weighted by $\lambda_c$, which balances the new context-aware objective against the standard forgetting term and the optional retention term. A larger $\lambda_c$ places more emphasis on contextual preservation. We study the effect of varying $\lambda_c$ on Gemma-2B-IT by evaluating six values, chosen based on the scale of each method's loss terms and spaced by doubling to ensure broad coverage.

Interestingly, we find that performance is largely insensitive to $\lambda_c$, making it easy to tune. As shown in Figure 9, multiple settings achieve near-optimal performance—matching the baseline in forgetting and overall utility, while substantially improving contextual utility toward the ideal level. For example, across all three methods, Contextual QA performance steadily increases as $\lambda_c$ grows: starting from degraded levels at $\lambda_c = 0$ (vanilla unlearning) and converging near the optimal range without decline. At the same time, Direct QA forgetting and model utility remain stable, with curves for different $\lambda_c$ values closely matching those of the original methods.

Since practitioners typically have access to both the forget and retain sets, they can directly assess forgetting, contextual utility, and overall utility to select the $\lambda_c$ that best fits their deployment goals. The robustness we observe across a wide range of $\lambda_c$ values makes our approach practical and simple to apply in deployments.

**Selecting $\lambda_c$.** For the context-aware results in the main text, we performed a grid search over six values of $\lambda$ (Figure 9). For each method, we identified the convergence epoch using the rule described earlier. We then select the one with the highest Contextual QA score (LLM-Judge) and model utility jointly among those that match the vanilla model's forgetting effectiveness—that is, Direct QA (LLM-Judge) within a tolerance $\delta$ of the vanilla baseline. Here, $\delta$ is the allowed slack in forgetting effectiveness to enable contextual improvements, which we set to 0.06 in our evaluation. That said, although we report the best choice, $\lambda$ is not highly sensitive (as shown in Figure 9); other values also work well with only slight variations or trade-offs across the three metrics.

## A.7. Additional Evaluation Metrics

To further validate our findings, we additionally evaluate Qwen3-8B using token-level accuracy and ChrF. As shown in Table 8, these metrics show trends consistent with our main ROUGE-L and LLM-Judge results: context-aware unlearning substantially improves Contextual QA, while maintaining comparable Direct QA performance.

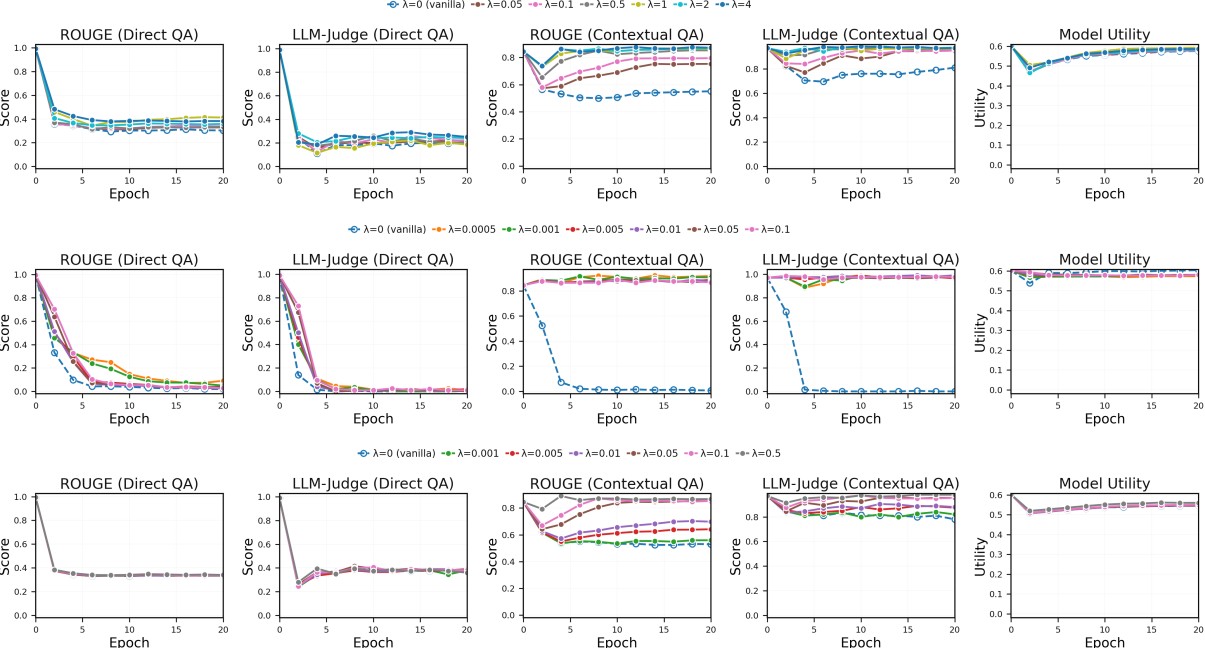

*Figure 9.* $\lambda$-ablation on the 5% forget set. Each row corresponds to one unlearning method (top to bottom: NPO, RMU, UNDIAL). Within each row, the subplots report Direct QA performance, Contextual QA performance, and Model Utility.

*Table 8.* Results on Qwen3-8B under the 5% forget set with additional metrics: token-level accuracy and ChrF.

| Method | Variant | TokAcc Direct ↓ | TokAcc Ctx. ↑ | ChrF Direct ↓ | ChrF Ctx. ↑ |
|---|---|---|---|---|---|
| NPO | Vanilla | 0.616 | 0.870 | 0.345 | 0.495 |
| | Context-aware | 0.643 | 0.951 (+0.081) | 0.345 | 0.647 (+0.152) |
| RMU | Vanilla | 0.053 | 0.080 | 0.040 | 0.037 |
| | Context-aware | 0.121 | 0.951 (+0.871) | 0.055 | 0.735 (+0.698) |
| UNDIAL | Vanilla | 0.390 | 0.886 | 0.421 | 0.591 |
| | Context-aware | 0.403 | 0.949 (+0.063) | 0.431 | 0.709 (+0.118) |

### A.8. Additional Baseline Results

We further evaluate context-aware unlearning on additional baselines using Gemma-2B-IT, including DPO, GradDiff, PDU (Entesari et al., 2025), and SimNPO (Fan et al., 2025). We exclude GradAscent because it collapses model utility nearly to zero and GradDiff can be viewed as its utility-preserving variant. As shown in Table 9, context-aware unlearning substantially improves Contextual QA across these methods while maintaining comparable forgetting effectiveness and model utility.

### A.9. Cost of the Reference Model

Our context-aware objective uses the original model as a frozen reference to compute the KL regularization term on contextual inputs. This follows the standard reference-model setup used in several unlearning and preference-optimization methods, such as DPO/NPO-style objectives. The reference model is not updated and introduces no additional trainable parameters.

In practice, the main extra cost is obtaining the reference output distribution for the contextual training examples. Since the reference model is fixed, these outputs can be precomputed and cached before unlearning, avoiding repeated reference-model forward passes during training. Thus, the additional cost can be largely shifted to a one-time preprocessing step.

*Table 9.* Additional baseline results on Gemma-2B-IT under the 5% forget set. We report ROUGE-L and LLM-Judge for Direct QA (↓) and Contextual QA (↑), along with model utility.

| Method | Variant | ROUGE-L | | LLM-Judge | | Utility |
|---|---|---|---|---|---|---|
| | | Direct ↓ | Context ↑ | Direct ↓ | Context ↑ | ↑ |
| DPO | Vanilla | 0.184 | 0.405 | 0.135 | 0.555 | 0.475 |
| | Context-aware | 0.237 | 0.749 (+0.344) | 0.190 | 0.855 (+0.300) | 0.483 |
| GradDiff | Vanilla | 0.005 | 0.001 | 0.000 | 0.000 | 0.422 |
| | Context-aware | 0.001 | 0.868 (+0.867) | 0.000 | 0.945 (+0.945) | 0.487 |
| PDU | Vanilla | 0.025 | 0.030 | 0.000 | 0.000 | 0.550 |
| | Context-aware | 0.041 | 0.924 (+0.894) | 0.000 | 0.975 (+0.975) | 0.549 |
| SimNPO | Vanilla | 0.469 | 0.706 | 0.375 | 0.940 | 0.578 |
| | Context-aware | 0.471 | 0.765 (+0.059) | 0.355 | 0.970 (+0.030) | 0.578 |

*Table 10.* Example of a long and noisy context. The gold fact is supported by the paragraph but not stated as a short direct answer.

| Type | Example |
|---|---|
| **Gold fact** | Hina Ameen primarily contributes to the geology genre. |
| **Noisy context** | Hina Ameen's recent publications in renowned scientific journals highlight her deep engagement with sedimentary processes and their implications for paleoenvironmental reconstruction. Her book, *Journey Through Earth's Layers*, meticulously explores the geological history of various regions, showcasing her expertise in field studies and mineral analysis. Through her lectures and workshops, she emphasizes ... |

## A.10. Robustness Under Longer and Noisy Contexts

In our main Contextual QA setting, the forgotten content is provided in a clean and direct form. This setting aligns with our primary evaluation scenario, where the forgotten material appears verbatim in the prompt—for example, a copyrighted passage removed from the model weights but reintroduced by a user at inference time. To complement this setting and provide a more comprehensive assessment, we additionally examine robustness under longer, noisier, and partially inconsistent contextual inputs.

Specifically, instead of supplying the original answer verbatim, we prompt GPT-4o-mini to produce a synthetic paragraph based on the answer. These paragraphs may include redundant, stylistic, or fabricated details, creating a more realistic yet noisy context. This generated text is then used as the contextual input. Table 10 shows a concrete example of this setting where the relevant fact must be inferred from a longer and less direct supporting passage.

On Gemma-2B-IT, we evaluate NPO, RMU, and UNDIAL along with their context-aware variants. All models are trained for 20 epochs to ensure convergence, and the hyperparameters of the context-aware variants are tuned so that their Direct QA forgetting and model utility closely match those of the vanilla methods. We report LLM-Judge scores for forgetting (↓) and model utility (↑) in Table 11.

*Table 11.* Long and noisy Contextual QA evaluation on Gemma-2B-IT. We report Direct QA LLM-Judge scores (↓), Contextual QA LLM-Judge scores (↑), and model utility (↑).

| Method | Variant | Direct LLM-Judge ↓ | Context LLM-Judge ↑ | Utility ↑ |
|---|---|---|---|---|
| NPO | Vanilla | 0.235 | 0.750 | 0.598 |
| | Context-aware | 0.255 | 0.850 (+0.100) | 0.598 |
| RMU | Vanilla | 0.015 | 0.010 | 0.599 |
| | Context-aware | 0.030 | 0.605 (+0.595) | 0.594 |
| UNDIAL | Vanilla | 0.330 | 0.695 | 0.538 |
| | Context-aware | 0.355 | 0.865 (+0.170) | 0.548 |

Even under long and noisy contexts, our main conclusion remains unchanged. Standard unlearning substantially reduces contextual QA performance, whereas context-aware unlearning consistently restores contextual behavior while maintaining comparable forgetting and utility. The improvement is particularly striking for RMU: its contextual LLM-Judge score rises from 0.010 to 0.605, showing that context-aware unlearning can recover strong contextual robustness even for methods that

fail almost completely in noisy settings.

## A.11. Mixed-Context Evaluation

To further assess robustness in contextual reasoning, we evaluate a more challenging mixed-context setting in which both the correct evidence and a conflicting distractor appear simultaneously in the prompt. Concretely, for each TOFU example, we inject the ground-truth supporting sentence alongside an answer candidate known to contradict the gold answer. These distractors are drawn from TOFU's conflict sets, which are lexically similar to the true answer but semantically incorrect. All experimental settings and model configurations follow Appendix A.10.

*Table 12.* Contextual QA when the context contains both the correct evidence and a conflicting distractor (Gemma-2B-IT).

| Method | Variant | Context ROUGE-L ↑ | Context LLM-Judge ↑ |
|--------|---------|-------------------|---------------------|
| NPO | Vanilla | 0.656 | 0.575 |
|  | Context-aware | 0.730 (+0.074) | 0.650 (+0.075) |
| RMU | Vanilla | 0.043 | 0.005 |
|  | Context-aware | 0.834 (+0.791) | 0.650 (+0.645) |
| UNDIAL | Vanilla | 0.495 | 0.600 |
|  | Context-aware | 0.853 (+0.358) | 0.845 (+0.245) |

As shown in Table 12, vanilla unlearning methods struggle substantially in the presence of conflicting evidence, often failing to identify the correct answer when a plausible distractor is present. RMU is especially brittle in this setting (0.043 ROUGE-L, 0.005 LLM-Judge). In contrast, context-aware variants yield large and consistent gains across all methods, in some cases improving performance by more than +0.79 ROUGE-L and +0.64 LLM-Judge. These results show that context-aware unlearning not only preserves contextual QA ability but also markedly enhances robustness when the model must disentangle correct information from conflicting distractors.

### A.11.1. EFFECT OF THE NUMBER OF DISTRACTORS IN CONTEXT

We further examine how the difficulty of the mixed-context setting scales as more conflicting distractor spans are introduced. For each TOFU example, we include the ground-truth evidence span and then insert $k \in \{2, 3, 4, 5\}$ distractor spans that contradict the gold answer. This setting evaluates whether the model can still identify and rely on the correct evidence as the amount of conflicting information becomes increasingly dominant.

*Table 13.* Effect of the number of conflicting distractors on Contextual QA LLM-Judge (↑) for Gemma-2B-IT. Context-aware rows include deltas relative to vanilla.

| Method | Variant | 2 distractors | 3 distractors | 4 distractors | 5 distractors |
|--------|---------|---------------|---------------|---------------|---------------|
| NPO | Vanilla | 0.620 | 0.595 | 0.560 | 0.575 |
|  | Context-aware | 0.670 (+0.050) | 0.605 (+0.010) | 0.620 (+0.060) | 0.620 (+0.045) |
| RMU | Vanilla | 0.000 | 0.000 | 0.005 | 0.005 |
|  | Context-aware | 0.540 (+0.540) | 0.450 (+0.450) | 0.360 (+0.355) | 0.345 (+0.340) |
| UNDIAL | Vanilla | 0.610 | 0.630 | 0.610 | 0.595 |
|  | Context-aware | 0.875 (+0.265) | 0.860 (+0.230) | 0.800 (+0.190) | 0.815 (+0.220) |

Table 13 reports Contextual QA performance (LLM-Judge) on Gemma-2B-IT. As the number of distractors increases, the overall contextual QA performance of all methods naturally declines due to the increased ambiguity introduced by multiple conflicting spans. However, the relative ordering remains consistent: context-aware variants substantially outperform their vanilla counterparts across all settings (e.g., +0.54 LLM-Judge for RMU at $k = 2$). These results indicate that context-aware unlearning markedly improves the model's ability to identify and rely on the correct evidence even when the context becomes increasingly cluttered with conflicting information.

