# OpenReview forum: "Forget to Know, Remember to Use: Context-Aware Unlearning for Large Language Models"
_ICML.cc/2026/Conference — ICML 2026 regular_

### Official Review · Reviewer_MB3G · 2026-02-21

**Soundness:** 3
**Presentation:** 4
**Significance:** 3
**Originality:** 4
**Overall Recommendation:** 6
**Confidence:** 3

**Summary:**

The paper asks if after unlearning, can a model still use the removed knowledge when a user explicitly provides it in the prompt? The answer, the authors show, is mostly no, and that is a problem.
The authors introduce contextual utility to name this capability and design a Contextual QA evaluation to measure it, the model receives the forgotten information as context and must answer a related question correctly. They test six unlearning methods on Gemma-2B-IT and Qwen3-8B using TOFU and find that all of them damage contextual utility, sometimes catastrophically, even when standard forgetting and retain-set metrics look fine.
The fix they propose is a KL-divergence term added to the existing unlearning objective. It keeps the unlearned model's output distribution on contextual queries close to the original model's. The term is modular in fact it plugs into RMU, NPO, and UNDIAL without touching their forgetting logic. With it, contextual performance recovers to near-perfect levels while forgetting and model utility stay essentially flat. The authors verify this across forget set sizes, a second dataset (PISTOL), and several harder context conditions including paraphrased and reasoning-based inputs.

**Compliance With Llm Reviewing Policy:**

Affirmed.

**Key Questions For Authors:**

1. Have you tested or do you expect the contextual degradation pattern to hold at 70B+ scale? Do you have evidence suggesting why it should or shouldn't?
2. Did you check agreement between the binary LLM-Judge and human judgments? The case studies show partially correct outputs, how often does the binary score misrepresent actual quality?
3. Appendix A.5 explains how you choose $\lambda_c$ and the convergence/stopping epoch. Were these choices made using a validation set that was fully separate from the final evaluation set? If not, could this make the reported gains look optimistic?

**Limitations:**

yes

**Strengths And Weaknesses:**

- Soundness: The core empirical finding is well-supported within the tested regime, and the experimental design is solid. A few points constrain the generality of the results. The experiments stay at 2B and 8B scale; larger models with stronger in-context learning abilities may resist contextual degradation differently, though the consistency of findings across two architecturally distinct models (Gemma, Qwen) provides some evidence of generality. The binary LLM-Judge offers no partial credit, which is a poor fit for the subtly wrong or partially hallucinated outputs the case studies reveal (e.g., NPO mixing correct and fabricated details). Without human-agreement calibration, the metric's reliability on edge cases is uncertain. On the method side, building $S_f^{\text{ctx}}$ from TOFU is mechanical since the forget set already contains question--answer pairs that can be reformatted into contextual prompts. While constructing $S_f^{\text{ctx}}$ from raw-text forget sets (e.g., copyright passages, personal documents) is feasible via standard RAG-style question generation, the paper does not discuss this step, leaving the practical effort and potential failure modes of the plug-in underspecified for realistic unlearning scenarios beyond QA-structured benchmarks.
- Presentation: The paper is clearly written and easy to follow. The case studies in Tables 1, 3, and 4 make the problem immediately concrete. Related work is positioned correctly. One minor structural issue: Section 3 presents diagnostic experiments on existing methods, Section 4 introduces the proposed fix, and Section 5 evaluates it, a reasonable problem→solution→evaluation flow. However, the overlap in experimental setup descriptions between Sections 3 and 5 creates some redundancy that could be streamlined.
- Significance: The problem is real. LLM deployment in RAG pipelines is used now days, and showing that unlearning can destroy contextual grounding is a practically relevant finding. The plugin nature of the fix makes adoption easy. The generalizability concern remains, but the paper partially addresses it via PISTOL (a more structurally entangled synthetic benchmark). Still, both benchmarks remain synthetic and limited relative to real GDPR/copyright unlearning workloads. The paper partially probes robustness beyond verbatim contexts, but evidence remains limited to synthetic/generated variants and small/medium models.
- Originality: The paper's clearest original contribution is identifying that parametric  forgetting and contextual usability are distinct properties that existing  methods conflate. Defining and measuring contextual utility as a separate evaluation dimension opens a useful new direction for the field. The novelty is therefore primarily diagnostic rather than technical.

---

> ### Author Rebuttal · Authors · 2026-03-31
>
> We thank the reviewer for recognizing the strengths of our work and for the thoughtful feedback. We address the main questions below.
>
> ### **Q1: Scaling**
> We agree that evaluating at 70B+ scale would be valuable. While such experiments are currently beyond our available compute budget, we expect the same unlearning degradation pattern (on contextual utility) to hold at larger scales because it appears to arise primarily from the unlearning objective rather than model scale. In particular, existing objectives tend to suppress responses associated with the forget set without conditioning on whether the same information is later reintroduced as valid external context, which can impair context-grounded answering regardless of architecture. The consistency of this pattern across two distinct model families (Gemma and Qwen) and across both TOFU and PISTOL provides initial evidence that the issue is not tied to a specific model class or benchmark.
>
> ### **Q2: LLM-Judge reliability**
> Thank you for raising this important point. To assess the reliability of the LLM-Judge metric, we randomly sampled 120 `(question, ground truth, response)` triples across all methods used in Sec. 5 and their context-aware variants and conducted human evaluation. For each triple, a human annotator assigned a binary label (1 for correct and aligned with the ground truth, 0 otherwise), which we compared against the binary outputs of LLM-Judge. We further performed bootstrap resampling (5,000 runs) to estimate 95% confidence intervals (CIs).
>
> | Metric | Value | 95% CI |
> | :--- | :---: | :---: |
> | **Agreement** | 0.9750 | [0.9417, 1.0000] |
> | **Cohen’s $\kappa$** | 0.9492 | [0.8835, 1.0000] |
> | **LLM Positive Rate** | 0.4500 | - |
> | **Human Positive Rate** | 0.4250 | - |
>
> These results show strong agreement between human evaluation and LLM-Judge ($\kappa = 0.9492$ with a tight CI), indicating that LLM-Judge provides a reliable proxy in our setting. We also agree that partially correct responses mixed with hallucinated details are a meaningful edge case for binary evaluation. In our factual QA setting, however, the intended criterion is whether the final answer is fully correct and reliable, since even partially correct answers can still be misleading in realistic use cases if they contain fabricated details. Importantly, human annotation and LLM-Judge largely agree on this decision rule, suggesting that this issue does not materially affect our conclusions.
>
> ### **Q3: Validation protocol**
> Thank you for raising this point. We would like to clarify that both Direct QA and Contextual QA are evaluated on the forget set (i.e., the direct training set during unlearning), since our goal is to measure forgetting behavior precisely on the target data to be unlearned. For the same reason, checkpoint selection is also based on forget-set performance, so the selection criterion is aligned with the unlearning objective itself rather than a separate downstream target. We will clarify this more explicitly in the revision.
>
> At the same time, our main conclusions do not rely on a single selected checkpoint: we also report results across training epochs (e.g., Fig. 4), which show consistent trends over time. This provides additional evidence that the observed gains are robust and not an artifact of checkpoint selection.
>
> ### **W1 & W3: Scope beyond QA-structured forget sets**
> We agree that extending the evaluation beyond TOFU-style forget sets that contain question-answer pairs is an important next step. Our current setup is a controlled first step designed to isolate and cleanly expose the unintended degradation of context-grounded answering after unlearning, while enabling systematic comparison across methods and settings using established benchmarks. As the reviewer notes, the same evaluation pipeline can be applied to raw-text forget sets by first constructing context-grounded question-answer instances from the target text (e.g., via standard LLM-based or RAG-style question generation). We also agree that extending the evaluation pipeline to more realistic unlearning settings, such as GDPR- or copyright-driven scenarios beyond synthetic benchmarks, is an important direction for future work.
>
> ### **W2: Presentation**
> Thank you for this suggestion. We will streamline the presentation by reducing overlapping experimental setup descriptions between Sec. 3 and Sec. 5.

---

> > ### Author Rebuttal · Reviewer_MB3G · 2026-03-31
> >
> > The rebuttal fully addresses all my concerns. I raise my score from 5 to 6 (Strong Accept).

---

> > > ### Author Response · Authors · 2026-04-06
> > >
> > > Thank you for the engagement with our rebuttal. We’re very glad the rebuttal fully addressed your concerns and appreciate your thoughtful assessment.

---

### Official Review · Reviewer_Ld3A · 2026-03-08

**Soundness:** 4
**Presentation:** 4
**Significance:** 4
**Originality:** 2
**Overall Recommendation:** 5
**Confidence:** 5

**Summary:**

This paper shows that existing LLM unlearning methods degrade the model's ability to use forgotten knowledge when it's re-provided as context at inference time. The authors propose a KL-divergence regularization term that preserves this "contextual utility" by aligning the unlearned model's contextual responses with the original model's. Experiments across multiple methods and models on the TOFU benchmark show the fix restores contextual QA to near-original levels without sacrificing forgetting or retain-set utility.

**Compliance With Llm Reviewing Policy:**

Affirmed.

**Final Justification:**

My concerns have been adequately addressed.

The added PDU and SimNPO results on Gemma are compelling — the jump in Contextual QA for PDU from near zero to 0.975 LLM-Judge, while preserving forgetting and utility, strongly supports the generality of the approach.

Overall, the paper identifies a gap in unlearning evaluation and proposes a simple, general fix backed by solid experiments. I'm maintaining my accept.

**Key Questions For Authors:**

See weaknesses.

**Limitations:**

Yes

**Strengths And Weaknesses:**

## Strengths

1. **Useful problem identification.** The paper points out a real gap in how unlearning methods are evaluated—existing work doesn't check whether the model can still use forgotten information when it shows up in context. This is a practical concern, especially with the rise of RAG pipelines, and the paper does a reasonable job establishing that the problem exists across multiple methods.

2. **The proposed fix is straightforward and general.** Adding a KL term to preserve contextual behavior is not a complex idea, but it works well and plugs into existing methods without much effort. The λ_c ablation (Figure 9) suggests it's not overly sensitive to tuning, which is a plus for adoption.

3. **Good qualitative examples.** The case studies (Tables 1, 3, 4) do a nice job showing the failure modes concretely—RMU degenerating into gibberish, NPO hallucinating, etc. These make the problem tangible in a way the aggregate numbers alone wouldn't.

## Weaknesses

1. **Limited benchmark scope.** The main evaluation relies heavily on TOFU, which consists of simple fictitious author QA pairs. The PISTOL results in the appendix help but are still narrow. The MUSE benchmark [1] evaluates unlearning over large-scale corpora (e.g., Harry Potter books, news articles). Its setup naturally involves longer, more realistic text where the tension between forgetting and contextual use would be more pronounced than in short QA pairs, making it a much better fit for validating the paper's central claims.

2. **Incomplete coverage of recent unlearning methods.** The baselines miss several relevant recent methods. Entesari et al. [2] propose a constrained optimization formulation with a logit-margin flattening loss evaluated on both TOFU and MUSE—directly relevant since their approach to the forgetting-retention trade-off may interact differently with contextual utility. Zhai et al. [3] propose token-selective entropy maximization restricted to sensitive prefixes and top-k logits, which avoids broad representational suppression and may naturally preserve contextual utility better than the methods tested here. SimNPO [4] is another relevant baseline that removes reference model bias in NPO-style unlearning. Testing on or comparing against these would strengthen the generality claims.
Finally, the work [5] seems extremely relevant, and they also try to address the issue of contextual information.

3. **The context-aware term requires the original model at training time.** The KL term aligns contextual responses to the frozen pre-unlearning model, so you need to store and run inference on the original model during unlearning. DPO-style methods already do this, but the paper doesn't discuss the memory/compute cost or how this scales to larger models.


## References

[1] Shi W, Lee J, Huang Y, Malladi S, Zhao J, Holtzman A, Liu D, Zettlemoyer L, Smith NA, Zhang C. Muse: Machine unlearning six-way evaluation for language models. arXiv preprint arXiv:2407.06460. 2024 Jul 8.

[2] Entesari T, Hatami A, Khaziev R, Ramakrishna A, Fazlyab M. Constrained Entropic Unlearning: A Primal-Dual Framework for Large Language Models. arXiv preprint arXiv:2506.05314. 2025 Jun 5.

[3] Zhai N, Shao P, Zheng B, Yang Y, Shen F, Bai L, Yang X. Maximizing Local Entropy Where It Matters: Prefix-Aware Localized LLM Unlearning. arXiv preprint arXiv:2601.03190. 2026 Jan 6.

[4] Fan C, Liu J, Lin L, Jia J, Zhang R, Mei S, Liu S. Simplicity prevails: Rethinking negative preference optimization for llm unlearning. arXiv preprint arXiv:2410.07163. 2024 Oct 9.

[5] Eldan R, Russinovich M. Who’s harry potter? approximate unlearning for LLMs.

---

> ### Author Rebuttal · Authors · 2026-03-31
>
> We thank the reviewer for the thoughtful feedback and for recognizing the value of our work. We address the main concerns below.
>
> ### **W1: Benchmark scope**
> Thank you for this thoughtful suggestion. We agree that evaluating in longer, more realistic text settings (e.g., MUSE) is important.
>
> **(1) Coverage of more realistic settings.**
> While our main results focus on TOFU and PISTOL, we do include evaluations that move in this direction. In Appendices A.6 and A.7, we test settings where supporting evidence is embedded in longer, noisy paragraphs or mixed with conflicting spans. These partially capture the challenges of contextual reasoning in more realistic text, and we will make this connection clearer.
>
> **(2) Role of synthetic benchmarks.**
> We use synthetic datasets (e.g., TOFU, PISTOL) deliberately because they provide a controlled setting to isolate the phenomenon of contextual degradation. By pairing synthetic facts with QA instances, they tightly couple target knowledge to individual samples, making both knowledge injection and unlearning cleanly controllable. Importantly, they also avoid confounding from pretraining memorization, enabling clearer attribution of effects and more direct comparisons across methods. This control is key to deriving clean, interpretable conclusions about contextual degradation.
>
> **(3) On using MUSE.**
> We agree that the MUSE benchmark is a valuable testbed for corpus-level unlearning. At the same time, it is not a drop-in replacement for our setting: MUSE is designed around large-scale corpus removal rather than fact-targeted QA-style evaluation. Adapting it to our evaluation protocol would require non-trivial dataset construction (e.g., mapping corpus-level deletions to QA-style queries with controlled context) and corresponding evaluation design. We view this as a worthwhile but more involved extension and will highlight it as an important direction for future work.
>
> ### **W2: Coverage of recent unlearning methods**
> Thank you for raising this point. In response, we added recent baselines on Gemma, including PDU [2] and SimNPO [4]. As shown below, the same trend still holds: our context-aware objective substantially improves Contextual QA while preserving similar Direct QA forgetting and utility.
>
> | Method | Direct Rouge ↓ | Direct LLMJ ↓ | Context Rouge ↑ | Context LLMJ ↑ | Utility |
> | :--- | :---: | :---: | :---: | :---: | :---: |
> | **PDU** | 0.025 | 0.000 | 0.030 | 0.000 | 0.550 |
> | **Context-Aware PDU** | 0.041 | 0.000 | 0.924 (+0.894) | 0.975 (+0.975) | 0.549 |
> | **SimNPO** | 0.469 | 0.375 | 0.706 | 0.940 | 0.578 |
> | **Context-Aware SimNPO** | 0.471 | 0.355 | 0.765 (+0.059) | 0.970 (+0.030) | 0.578 |
>
> In particular, for PDU, our method improves contextual performance from nearly zero to 0.975 LLM-Judge score, while maintaining comparable forgetting and utility. This further supports the generality of our approach.
>
> Regarding [3], thank you for bringing it to our attention. This work appeared very recently (roughly three weeks before the ICML deadline), and we did not find a public implementation available. As a result, we were not able to conduct a faithful comparison within the rebuttal window. We will discuss it in the revision and leave comparison to future work.
>
> Thank you for pointing out [5]. While relevant, [5] has a fundamentally different goal and emphasis. It focuses on the standard unlearning setting, aiming to suppress the generation of target (e.g., copyrighted) content by steering the model toward generic continuations when sensitive target concepts appear in the input context. In contrast, our setting focuses on preserving the use of forgotten content; specifically, ensuring the model retains the ability to correctly utilize unlearned knowledge when it is explicitly provided in an external context. We will discuss and clarify this critical distinction in the related work section of the final version.
>
> ### **W3: Cost of the KL term/limitation discussion**
> We agree that the cost of requiring the original model should be discussed more explicitly. Our method assumes access to a frozen reference model during unlearning, since the $KL$ term aligns context-conditioned outputs to the pre-unlearning model. This is a standard assumption in reference-based unlearning methods, including DPO/NPO-style approaches and RMU, and therefore does not introduce a fundamentally new requirement beyond this class of methods.
>
> We also agree that scalability to larger models is important. Because the reference model is completely frozen, its outputs on the contextual training set can be precomputed and cached, which can reduce repeated reference-model forward passes during training. We will clarify this scalability consideration in the revision.

---

> > ### Author Rebuttal · Reviewer_Ld3A · 2026-04-03
> >
> > Thank you for your response and further experiments. They further showcase the value of adding a context-aware loss term.

---

> > > ### Author Response · Authors · 2026-04-06
> > >
> > > We’re glad the response and additional experiments addressed your concerns and will include them in the final version. Thank you for the follow-up.

---

### Official Review · Reviewer_wwdJ · 2026-03-10

**Soundness:** 3
**Presentation:** 3
**Significance:** 3
**Originality:** 3
**Overall Recommendation:** 5
**Confidence:** 3

**Summary:**

This paper studies an important issue in the evaluation of LLM unlearning: after a model has unlearned target knowledge, it may still be desirable for the model to use that knowledge correctly when it is explicitly reintroduced in the input context. The paper refers to this ability as contextual utility.

To measure it, the authors introduce a new evaluation setting Contextual QA and show that several existing unlearning methods perform poorly under this setting. To address this issue, the paper proposes a simple context-aware objective based on KL consistency, which can be added to existing unlearning objectives to preserve the model's contextual behavior after unlearning.

**Compliance With Llm Reviewing Policy:**

Affirmed.

**Final Justification:**

The rebuttal addressed my concerns. I have raised my score from 4 (Weak accept) to 5 (Accept).

**Key Questions For Authors:**

1. It remains unclear whether the proposed approach can effectively generalize to weaker baselines. Could the authors show the effectiveness of the proposed objective on these weaker methods to provide a more comprehensive assessment of its robustness?

2. The current Contextual QA setting is still somewhat idealized, since the gold fact is directly provided in the prompt. Although Appendix A.6 considers longer and noisier contexts, I would appreciate seeing at least one concrete example of such noisy context to better understand how realistic this setting is.

    Additionally, the generalizability of the findings remains an open question in more complex settings. Exploring whether these results hold consistent in long-context or RAG scenarios (e.g., multi-document contexts or conversational contexts) would significantly strengthen the paper’s practical implications.

**Limitations:**

yes

**Strengths And Weaknesses:**

This paper has several clear strengths.

1. The problem formulation is interesting and practically meaningful.  The idea of evaluating contextual utility is useful, especially for realistic settings where forgotten information may later be reintroduced through user input or retrieval.
2. The proposed method is simple and easy to integrate into existing unlearning methods. The gains on Contextual QA are clear on both Gemma-2B-IT and Qwen3-8B, especially for RMU.
3. The paper is clearly written, and the additional experiments in the appendix addresses several concerns beyond the main paper.

My main concern is about baseline coverage.

While Figure 2 evaluates six unlearning baselines, the context-aware variant is only tested on three of them. The authors justify this by selecting the stronger methods, but these are also the methods that already perform relatively better on Contextual QA (in Figure 2). This makes it harder to judge how general the proposed approach is, especially for weaker baselines that degrade more severely.

---

> ### Author Rebuttal · Authors · 2026-03-31
>
> We thank the reviewer for the positive assessment and constructive feedback. We address the concerns below.
>
> ### **Q1: Generalization to weaker baselines**
> We agree that broader baseline coverage is valuable. In response, we further evaluate the proposed objective on additional weaker baselines on Gemma, including DPO and GradDiff. We exclude GradAscent as it collapses utility to nearly zero; GradDiff can be viewed as its utility-preserving variant.
>
> | Method | Direct Rouge ↓ | Direct LLMJ ↓ | Context Rouge ↑ | Context LLMJ ↑ | Utility |
> | :--- | :---: | :---: | :---: | :---: | :---: |
> | **DPO** | 0.184 | 0.135 | 0.405 | 0.555 | 0.475 |
> | **Context-Aware DPO** | 0.237 | 0.190 | 0.749 (+0.344) | 0.855 (+0.300) | 0.483 |
> | **GradDiff** | 0.005 | 0.000 | 0.001 | 0.000 | 0.422 |
> | **Context-Aware GradDiff** | 0.001 | 0.000 | 0.868 (+0.867) | 0.945 (+0.945) | 0.487 |
>
> We observe that for GradDiff, context-aware unlearning improves contextual performance from nearly zero to 0.945 LLM-Judge Score, while maintaining similarly strong forgetting and comparable utility. For DPO, direct QA performance increases slightly, accompanied by a larger gain in contextual performance and essentially unchanged utility. These results show that our method remains effective beyond the stronger baselines considered in the main paper, further supporting its generality.
>
> ### **Q2: Longer, Noisier Contexts**
> Thank you for the suggestion. As you noted, Appendix A.6 explores longer and noisier contexts. We provide a concrete example here to better illustrate this setting.
>
> | Type | Example |
> | :--- | :--- |
> | **Gold fact** | Hina Ameen primarily contributes to the geology genre. |
> | **Noisy context** | Hina Ameen's recent publications in renowned scientific journals highlight her deep engagement with sedimentary processes and their implications for paleoenvironmental reconstruction. Her book, "Journey Through Earth’s Layers," meticulously explores the geological history of various regions, showcasing her expertise in field studies and mineral analysis. Through her lectures and workshops, she emphasizes... |
>
> In this setting, the model must infer the relevant fact from a longer, indirect description. This better reflects realistic inputs such as retrieved passages or user-provided background context.
>
> Our current setup is a first step designed to cleanly expose the unintentional spillover effect of unlearning, but it is naturally extensible to more complex scenarios. We agree that broader scenarios, such as long-context, multi-document, or conversational settings, are important directions for understanding the broader practical implications of contextual utility. We will clarify this motivation in the revision and leave these broader settings to future work.

---

> > ### Author Rebuttal · Reviewer_wwdJ · 2026-04-02
> >
> > Thank you for the response and new results, which address my concerns. I appreciate the discussion on future directions and will raise my score to 5.

---

> > > ### Author Response · Authors · 2026-04-06
> > >
> > > Thank you for your update and thoughtful evaluation. We’re glad the response addressed your concerns.

---

### Official Review · Reviewer_bvfj · 2026-03-11

**Soundness:** 3
**Presentation:** 2
**Significance:** 3
**Originality:** 3
**Overall Recommendation:** 4
**Confidence:** 5

**Summary:**

This paper identifies that existing LLM unlearning methods, while effectively removing targeted knowledge, severely degrade contextual utility—the ability to correctly utilize "forgotten" information when it is explicitly provided in the input context. The authors propose Contextual QA as a new evaluation protocol and introduce "Context-aware" which is a KL-divergence regularization enhancement that aligns unlearned models with the original model on contextual queries. Experiments on Gemma-2B-IT and Qwen-3-8B demonstrate that this approach restores contextual utility to near-perfect levels without sacrificing forgetting effectiveness or retain-set performance, providing a practical solution to prevent "over-forgetting" in real-world deployments.

**Compliance With Llm Reviewing Policy:**

Affirmed.

**Key Questions For Authors:**

Q1: Can you provide evaluation of the proposed method using additional metrics beyond ROUGE-L and LLM-Judge (e.g., perplexity, ChrF, or token-level accuracy)?

Q2: Can you provide mechanistic insights into why existing unlearning methods degrade contextual utility and why KL-regularization specifically mitigates this?

Q3: Table Clarity and Limitations Disclosure. (1) Would you revise the table headers for Table 3 and Table 4 to clearly distinguish their respective analytical purposes? (2) Crucially, would you add a dedicated Limitations section or expand the discussion of the "forgetting gap"—specifically addressing how practitioners should balance the trade-off between restored contextual utility and the slightly reduced forgetting effectiveness observed in your results?

If the authors provide satisfactory responses to these questions, I will increase my rating.

**Limitations:**

No

Missing Limitations Discussion. Notably, the manuscript lacks a dedicated limitations paragraph. As shown in the experimental results, the context-aware approach does incur minor trade-offs in unlearning effectiveness (e.g., slight increases in forget-set performance compared to vanilla methods). Authors should be rewarded rather than punished for being up front about the limitations of their work and any potential negative societal impact; explicitly discussing these trade-offs would strengthen the paper's credibility.

**Strengths And Weaknesses:**

Strengths:
1. Novel Problem Formulation: The paper identifies a critical gap in existing unlearning evaluations by introducing Contextual QA, which captures practical RAG scenarios where models must process "forgotten" knowledge provided in context. This reveals that current methods suffer from "over-forgetting"—suppressing not just memorization but the ability to utilize contextual information.
2. Simple yet Effective Solution: The proposed KL-divergence regularization is elegant and plug-and-play, requiring minimal modification to existing unlearning objectives. Despite its simplicity, it achieves near-perfect restoration of contextual utility without compromising forgetting effectiveness or retain-set performance.
3. Comprehensive Empirical Validation: The experiments cover three state-of-the-art unlearning methods across two popular instruction-tuned models (Gemma-2B-IT and Qwen-3-8B), demonstrating the method's generalizability and robustness.

Weaknesses:
1. Limited Evaluation Metrics: The quantitative assessment relies primarily on only two metrics—ROUGE-L and LLM-Judge—which may provide an incomplete picture of model behavior. The inclusion of complementary metrics such as ChrF, perplexity, or token-level factual accuracy would offer a more comprehensive evaluation across different dimensions (e.g., lexical diversity, fluency, and semantic preservation).
2. Ambiguous Table Organization: The paper presents Table 3 and Table 4 with identical headers despite their distinct analytical purposes—Table 3 compares different unlearning methods while Table 4 analyzes performance across datasets. This inconsistent presentation is confusing and reduces readability; distinct headers reflecting their specific analytical perspectives would significantly improve clarity.
3. Insufficient Mechanistic Analysis: While the paper effectively demonstrates that context-aware unlearning works (via performance curves and case studies), it lacks in-depth analysis of why existing methods degrade contextual utility and why KL regularization successfully mitigates this. The absence of mechanistic insights—such as attention visualization, or representation probing on the KL term's specific impact—limits the work's theoretical depth and its ability to provide actionable guidance for future research in this domain.

---

> ### Author Rebuttal · Authors · 2026-03-31
>
> We thank the reviewer for the positive assessment and constructive suggestions. We address each point below.
>
> ### **Q1 & W1: Additional evaluation metrics**
> Thanks for suggesting these complementary metrics. We evaluate Qwen3-8B using ChrF and token-level accuracy for NPO, RMU, UNDIAL, and their corresponding context-aware variants. We observe a consistent trend with our main results: context-aware unlearning maintains comparable (or slightly higher) performance on Direct QA while substantially improving Contextual QA.
>
> | Method | Variant | TokAcc Direct ↓ | TokAcc Context ↑ | ChrF Direct ↓ | ChrF Context ↑ |
> | :--- | :--- | :---: | :---: | :---: | :---: |
> | **NPO** | Vanilla | 0.616 | 0.870 | 0.345 | 0.495 |
> | **NPO** | Context-aware | 0.643 | 0.951 (+0.081) | 0.345 | 0.647 (+0.152) |
> | **RMU** | Vanilla | 0.053 | 0.080 | 0.040 | 0.037 |
> | **RMU** | Context-aware | 0.121 | 0.951 (+0.871) | 0.055 | 0.735 (+0.698) |
> | **UNDIAL** | Vanilla | 0.390 | 0.886 | 0.421 | 0.591 |
> | **UNDIAL** | Context-aware | 0.403 | 0.949 (+0.063) | 0.431 | 0.709 (+0.118) |
>
> We believe these results further strengthen our main claim that the proposed objective specifically restores context-conditioned utility.
>
> For perplexity (PPL), we found it less informative for this fact-sensitive QA setting. In particular, some vanilla unlearning models (e.g., RMU) may produce repetitive strings with artificially low fluency PPL, despite clearly poor answer quality. Thus, PPL does not reliably reflect contextual correctness here. We will add the new results and clarify the limitation of PPL in the revision.
>
> ### **Q2 & W3: Mechanistic insight**
> **(1) Why existing unlearning can affect contextual use of information.**
> Our intuition is that most unlearning objectives only reward removing the undesired response on the forget set, without constraining the extent of suppression. As long as forgetting improves, broader suppression is not penalized and may even be favored. For example, in methods like RMU, unlearning perturbs hidden representations induced by target-domain inputs rather than only suppressing specific outputs. Consequently, semantically related content—when later provided as external context—can be mapped to the same distorted regions of representation space, leading to degraded or abnormal contextual use.
>
> To address this, our method introduces a $KL$ regularization term that encourages the model to preserve correct conditional behavior when supporting evidence is explicitly provided. Empirically, this improves contextual performance while maintaining comparable forgetting and overall utility.
>
> **(2) On mechanistic insights (e.g., attention visualization).**
> We agree that mechanistic analysis is valuable. However, tools like attention visualization can be difficult to interpret in this setting and often require careful, task-specific design to yield reliable conclusions. A comprehensive understanding of how unlearning reshapes internal representations remains an open challenge for this field. We view our empirical results as a strong starting point and will emphasize deeper mechanistic investigation as an important direction for future work.
>
> ### **Q3 & W2: Table clarity and limitations/trade-off discussion**
> Thank you for pointing this out. We will improve the headers of Tables 3 and 4 to be clearer. We clarify that, while both tables compare vanilla vs. context-aware unlearning, they emphasize different axes: Table 3 compares across unlearning methods, while Table 4 compares across context variants. We will revise the headers to make this distinction more explicit.
>
> We also agree that limitations and trade-offs should be discussed more directly and will add a dedicated *Limitations* section. Empirically, context-aware unlearning yields large gains in Contextual QA, in some cases nearly restoring full contextual utility, and in some cases without degrading forgetting performance (e.g., UNDIAL in Table 2). However, there is still a trade-off: in some settings, substantial gains in contextual utility may come with a small reduction in Direct QA forgetting. The preferred balance depends on deployment priorities, and we will clarify this in the final version.

---

> > ### Author Rebuttal · Reviewer_bvfj · 2026-04-07
> >
> > I have carefully read the authors' rebuttal. I appreciate the authors' efforts in Q1 and Q3. But the authors have not addressed my core concerns (Q2).
> >
> > The manuscript lacks principled and mechanistic insights into why the proposed loss function works. The current presentation primarily relies on empirical performance gains to "demonstrate" its effectiveness, without providing deeper theoretical or interpretability analysis.
> >
> > I believe there to be fundamental flaws with the work. Please see the above comment.

---

> > > ### Author Response · Authors · 2026-04-08
> > >
> > > We thank the reviewer for the follow-up. We agree that mechanistic understanding is important. At the same time, existing unlearning methods vary substantially and rely on different mechanisms (e.g., stochastic perturbations to activations or preference-based objectives such as preference reversal), making principled attribution to specific components inherently challenging. As a result, the interpretability of unlearning remains an open problem. Our main contribution in this work is to identify a previously overlooked issue, i.e., contextual degradation after unlearning, and to provide a simple and effective empirical solution. Our claims are therefore grounded in consistent empirical evidence across models and methods. We will explicitly address this in the limitations section and highlight mechanistic analysis as an important direction for future work.

---

### Decision · Program_Chairs · 2026-04-30

**Decision:**

Accept (regular)

**Comment:**

This paper introduces an important evaluation gap of LLM unlearning, called “contextual utility”: after a model has unlearned target knowledge, it may still be desirable for the model to use that knowledge correctly when it is explicitly reintroduced in the input context. The authors introduce a new evaluation setting, Contextual QA, and show that several existing unlearning methods perform poorly under this setting. To address this issue, the paper proposes a simple context-aware objective based on KL consistency, which can be added to existing unlearning objectives to preserve the model's contextual behavior after unlearning.

There was a general agreement among reviewers that contextual utility is a substantial gap in unlearning evaluation, that the proposed approach is simple and generally applicable, and that the writing is clear. There were some issues regarding missing results and baselines, and other less critical points – most of these points seem to have been resolved during the rebuttal. Reviewer bvfj also expected the paper to include mechanistic insights. While I agree that such insights would add depth, I agree with the authors’ point that they are not necessary for supporting their arguments.

Overall, the paper highlights an evaluation gap of unlearning methods that has been overlooked in prior studies. It contributes the Contextual QA evaluation setup and an effective, practical method to mitigate this gap. I believe this work makes valuable contributions to the community.